# Show-o2: Improved Native Unified Multimodal Models

Jinheng Xie[1]   Zhenheng Yang[2]   Mike Zheng Shou[1*]

[1] Show Lab, National University of Singapore   [2] ByteDance

## Abstract

This paper presents improved native unified multimodal models, *i.e.,* Show-o2, that leverage autoregressive modeling and flow matching. Built upon a 3D causal variational autoencoder space, unified visual representations are constructed through a dual-path of spatial (-temporal) fusion, enabling scalability across image and video modalities while ensuring effective multimodal understanding and generation. Based on a language model, autoregressive modeling and flow matching are natively applied to the language head and flow head, respectively, to facilitate text token prediction and image/video generation. A two-stage training recipe is designed to effectively learn and scale to larger models. The resulting Show-o2 models demonstrate versatility in handling a wide range of multimodal understanding and generation tasks across diverse modalities, including text, images, and videos. Code and models are released at https://github.com/showlab/Show-o.

## 1 Introduction

Large language models (LLMs) [108, 134] have achieved unprecedented performance levels, fueled by extensive web-scale text resources, substantial computational power, and billions of parameters. In the multimodal domain, large multimodal models (LMMs) [7, 29, 56] and visual generative models [37, 89, 126], have also demonstrated exceptional capabilities in tasks such as general-purpose visual question answering and text-to-image/video generation. Given their success, unified multimodal models (UMMs) [103, 121, 129] have been investigated to unify multimodal understanding and generation within a single model or system. In addition to multimodal understanding capability, this line of approaches seeks to simultaneously cultivate multimodal understanding and generation abilities in the model/system through pre-training, fine-tuning, or connecting tailored models.

Here, we provide a comparative analysis of selected UMMs in Table 1, focusing on two perspectives, including i) visual representations for understanding and generation and ii) the type of unified modeling. Generally, there are two approaches to incorporating visual representations for multimodal understanding and generation: i) a unified representation for both understanding and generation, as seen in works like Chameleon [103], Transfusion [148], and Show-o [129]; and ii) decoupled representations, utilizing CLIP [91] for multimodal understanding and variational autoencoder (VAE) for visual generation. To involve both multimodal understanding and generation capabilities, two primary methods have been explored: i) natively applying multimodal understanding and generation objectives within a single model and ii) tuning adapters to assemble tailored models. We refer the first type as *native unified multimodal models*, distinguishing it from the second type that assembles tailored models. These principles, combined with autoregressive or diffusion modeling or both, contribute to the development of unified multimodal models.

Compared to existing UMMs that primarily focus on text and image, our approach explores model designs that provide substantial potential and scalability in natively unifying text, image, and video modalities. An overview of our approach is presented in Fig. 1. Specifically, for visual inputs, we

---

* Corresponding Author

39th Conference on Neural Information Processing Systems (NeurIPS 2025).

Table 1: Comparative analysis of selected unified multimodal models based on the type of visual representations and unified modeling for multimodal understanding and generation. In this context, **native und. & gen.** refers to the direct decoding of output sequences into texts, images, and videos, as opposed to serving as conditions for decoding using external pre-trained decoders like Stable Diffusion. * indicates the method adopts two distinct models for multimodal understanding and generation, respectively. Diff. means the diffusion modeling. *Please refer to the complete Table 15 in the appendix.*

| Methods | Und. & Gen. Representation | | | Type of Unified Modeling | | |
|---|---|---|---|---|---|---|
| | Unified | Decoupled | Support Video | Native Und. & Gen. | Assembling Tailored Models | Paradigm |
| Chameleon [103] | ✓ | | ✗ | ✓ | | AR |
| Transfusion [148] | ✓ | | ✗ | ✓ | | AR + Diff. |
| Show-o [129] | ✓ | | ✗ | ✓ | | AR + Diff. |
| VILA-U [124] | ✓ | | ✓ | ✓ | | AR |
| Emu3 [115] | ✓ | | ✓ | ✓ | | AR |
| LlamaFusion [96] | ✓ | | ✗ | ✓ | | AR + Diff. |
| Show-o2 (Ours) | ✓ | | ✓ | ✓ | | AR + Diff. |
| Janus-Series [26, 27, 80] | | ✓ | ✗ | ✓ | | AR (+Diff) |
| UnidFluid [38] | | ✓ | ✗ | ✓ | | AR + MAR |
| Mogao [66] | | ✓ | ✗ | ✓ | | AR + Diff. |
| BAGEL [32] | | ✓ | ✓ | ✓ | | AR + Diff. |
| NExT-GPT [121] | | ✓ | ✓ | | ✓ | AR + Diff. |
| SEED-X [40] | | ✓ | ✗ | | ✓ | AR + Diff. |
| ILLUME [112] | | ✓ | ✗ | | ✓ | AR + Diff. |
| MetaMorph [107] | | ✓ | ✗ | | ✓ | AR + Diff. |
| MetaQueries [84] | | ✓ | ✗ | | ✓ | AR + Diff. |
| TokenFlow* [90] | ✓ | | ✗ | | ✓ | AR |

operate within the 3D causal VAE [109] space, which is capable of accommodating both images and videos. Recognizing the distinct feature dependencies between multimodal understanding and generation, we construct unified visual representations that simultaneously capture rich semantic information and low-level features with intrinsic structures and textual details from the visual latents. This is achieved through a dual-path mechanism consisting of semantic layers, a projector, and a spatial (-temporal) fusion process. As the fusion process occurs within the 3D causal VAE space, when it comes to videos, semantic and low-level features are temporally aligned and fused with full-frame video information.

Text embeddings and unified visual representations are structured into a sequence to go through a pre-trained language model and are modeled by a specific language head and flow head, respectively. Specifically, autoregressive modeling with causal attention is performed on the language head when dealing with text token prediction, and flow matching with full attention is applied to the flow head for image/video generation. Since the base language model lacks visual generation capabilities, we propose a two-stage training recipe to effectively learn such an ability while retaining the language knowledge, without requiring a massive text corpus. In the first stage, we mainly focus on pre-training the flow head for visual generation using (interleaved) text, image, and video data. In the second stage, the full model is fine-tuned with high-quality multimodal understanding and generation data.

Extensive experimental results have demonstrated that our model surpasses the existing methods in terms of most metrics across multimodal understanding and visual generation benchmarks. Collectively, the main contributions of this paper can be summarized as:

- We present an improved native unified multimodal model that seamlessly integrates autoregressive modeling and flow matching, enabling a wide range of multimodal understanding and generation across (interleaved) text, images, and videos.

- Based on the 3D causal VAE space, we construct unified visual representations scalable to both multimodal understanding and generation, image and video modalities by combining semantic and low-level features through a dual-path of spatial (-temporal) fusion mechanism.

- We design a two-stage training pipeline that effectively and efficiently learns unified multimodal models, retaining language knowledge and enabling effective scaling up to larger models, without requiring a massive text corpus.

- The proposed model demonstrates state-of-the-art performance on multimodal understanding and visual generation benchmarks, surpassing existing methods across various metrics.

## 2 Related Work

### 2.1 Large Multimodal Models

Building upon the advancements of large language models (LLMs) [108, 134], large multimodal models (LMMs) [7, 29, 56, 61, 73] have showcased remarkable capabilities in general-purpose visual question answering. These approaches typically leverage pre-trained vision encoders to project visual features and align them within the embedding space of LLMs. Meanwhile, a growing number of encoder-free LMMs [34, 35, 129] aim to directly align raw visual features within the LLM embedding space. However, these encoder-free methods often fall behind models that utilize image-text-aligned visual features in terms of performance. Beyond model architecture, recent studies [20, 56, 106] have highlighted the critical role of high-quality instructional data in enhancing multimodal capabilities.

### 2.2 Visual Generative Models

Two prominent paradigms for visual generation, namely diffusion [9, 19, 71, 87, 89, 93, 94, 120, 126, 128, 141] and autoregressive modeling [22, 52, 59, 85, 98], have been extensively studied in image and video generation in recent years. Diffusion-based methods typically employ optimized architectures that integrate pre-trained text encoders with denoising networks. In contrast, autoregressive methods often utilize LLM-based architectures and are trained through next-token prediction. Recently, several studies [38, 62, 74] have explored hybrid approaches that combine diffusion and autoregressive modeling to further advance visual generation capabilities.

### 2.3 Unified Multimodal Models

Building on the success of large multimodal and visual generative models, pioneering unified multimodal models (UMMs) such as Chameleon [103], Show-o [129], and Transfusion [148] aim to integrate these capabilities into a single model through autoregressive or diffusion modeling or both. Further advancements [28, 50, 78, 97, 115, 124] have focused on optimizing the training pipeline and enhancing the semantics of discrete tokens, leading to improved performance. We refer to these approaches as *native unified multimodal models*, as they inherently combine multimodal understanding and generation objectives within a unified architecture.

An alternative and promising direction [18, 36, 40, 77, 84, 102, 107] for unifying multimodal understanding and generation involves assembling off-the-shelf specialized LMMs and visual generative models by tuning adapters or learnable tokens. Representative works [18, 40, 84, 121] have demonstrated the promising capabilities and intriguing properties of such assembled unified frameworks, highlighting their potential for further exploration.

## 3 Methodology

In this section, we introduce the overall framework (Section 3.1), which consists of two key components: i) the design of unified visual representations for multimodal understanding and generation, applicable to both images and videos, and ii) the native learning of multimodal understanding and generation capabilities. Subsequently, we present a two-stage training recipe (Section 3.2), which is designed to progressively learn and effectively scale up the unified multimodal model.

### 3.1 Overall Framework

**Overall Architecture.** An overview of our proposed unified model is depicted in Fig. 1. Given (interleaved) texts, images, or videos, a text tokenizer with an embedding layer and a 3D causal VAE encoder accordingly process them into continuous text embeddings and visual latent representations. Subsequently, the visual latent representations undergo a dual-path extraction of spatial (-temporal) fusion to create the unified visual representations. These representations are then structured into a sequence, which is fed into a language model equipped with language and flow heads to model the sequence via autoregressive modeling and flow matching accordingly. Finally, a text de-tokenizer in conjunction with a 3D causal VAE decoder is employed to decode the final output. Next, we will delve into the fundamental design principles behind the unified visual representation and flow head.

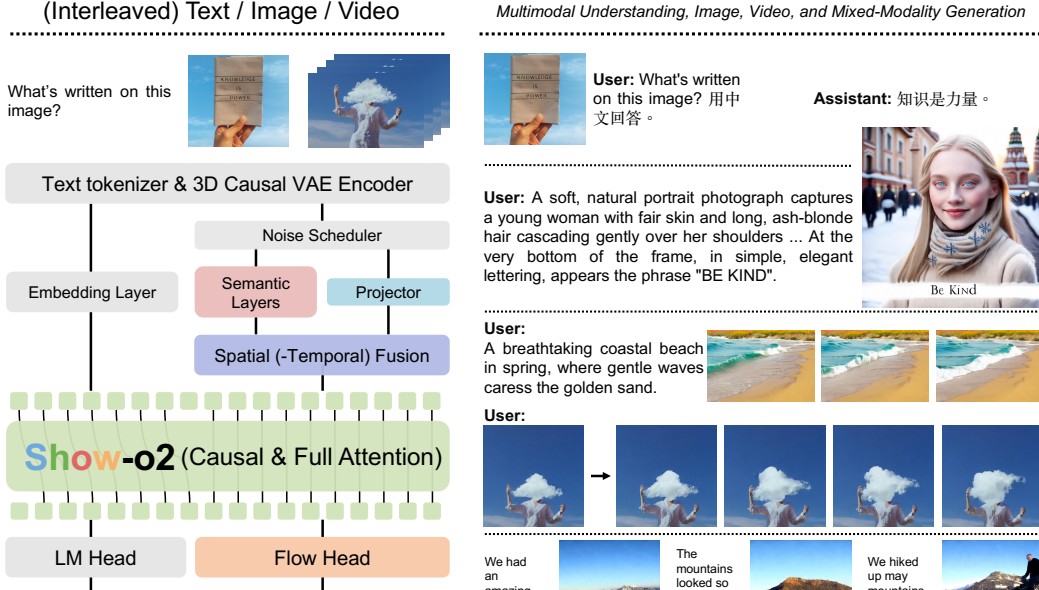

Figure 1: Our approach begins by encoding input texts, images, and videos into continuous embeddings and visual latents. The visual latents are processed through a dual-path extraction and spatial (-temporal) fusion mechanism to construct unified visual representations that are scalable for both multimodal understanding and generation, image and video modalities. These text embeddings and unified visual representations are then structured into a sequence for the base language model, equipped with dedicated heads. Specifically, text tokens are modeled autoregressively by a language head, while image and video latents are handled by a flow head using flow matching. We employ the omni-attention mechanism [129, 148] to enable causal attention along the sequence while maintaining full attention within the unified visual representations. This design empowers our model to effectively tackle tasks such as image/video understanding, generation, and mixed-modality generation.

**Unified Visual Representation.** To scalably support image and video modalities, we employ a 3D causal VAE encoder to extract image/video latents. As multimodal understanding and generation differ in feature dependency, we propose a dual-path architecture comprising semantic layers $\mathcal{S}(\cdot)$ to extract high-level representations of rich semantic contextual information and a projector $\mathcal{P}(\cdot)$ to retain complete low-level information from the extracted visual latents. Specifically, semantic layers $\mathcal{S}(\cdot)$ share the same vision transformer blocks of SigLIP [139] with a new $2 \times 2$ patch embedding layer. Given $n$ visual latents $\mathbf{x}_t = \{x_i\}_{i=1}^n$ at a noise level:

$$\mathbf{x}_t = t \cdot \mathbf{x}_1 + (1 - t) \cdot \mathbf{x}_0, \tag{1}$$

where $\mathbf{x}_0 \sim \mathcal{N}(0, 1)$ and $t \sim [0, 1]$, we load the pre-trained weights of SigLIP and pre-distill $\mathcal{S}(\cdot)$ as follows:

$$\mathcal{L}_{\text{distill}} = -\frac{1}{n} \sum \log \text{sim}(\mathcal{S}(\mathbf{x}_t), \texttt{SigLIP}(\mathbf{X})), \tag{2}$$

where $\mathbf{X}$ is the input image, $\texttt{SigLIP}(\cdot)$ extracts the image patch features, and $\text{sim}(\cdot)$ indicates the cosine similarity calculator. In this way, semantic layers $\mathcal{S}(\cdot)$ can mimic extracting semantic features from both clean and noised visual latents $\mathbf{x}_t$. The projector $\mathcal{P}(\cdot)$ is simply composed of a 2D patch embedding layer. The extracted high- and low-level representations are spatially (and temporally when it comes to videos) fused by concatenating through the feature dimension and applying RMSNorm [140] with two MLP layers to get the unified visual representations $\mathbf{u}$:

$$\mathbf{u} = \texttt{STF}(\mathcal{S}(\mathbf{x}_t), \mathcal{P}(\mathbf{x}_t)), \tag{3}$$

where $\texttt{STF}$ indicates the spatial (-temporal) fusion mechanism. In addition, we prepend a time step $t$ embedding to the unified visual representations for generative modeling. $t$ is set as 1.0 to get time step embedding for the clean image.

We structure the text embeddings and unified visual representations into a sequence following a general interleaved image-text format below:

$$[\text{BOS}] \ \{\text{Text}\} \ [\text{BOI / BOV}] \ \{\text{Image / Video}\} \ [\text{EOI / EOV}] \ \{\text{Text}\} \cdots [\text{EOS}].$$

The sequence format above is flexible and can be adapted to various input types. We adopt the omni-attention mechanism [129, 148] to let the sequence modeling be causal but with full attention within the unified visual representations.

**Flow Head.** Apart from the language head for text token prediction, we employ a flow head to predict the defined velocity $\mathbf{v}_t = \frac{d\mathbf{x}_t}{dt}$ via flow matching [71, 75]. Specifically, the flow head simply consists of several transformer layers with time step modulation via the adaLN-Zero blocks, as seen in DiT [86].

During training, we natively apply next token prediction $\mathcal{L}_{\text{NTP}}$ to the language head and flow matching $\mathcal{L}_{\text{FM}}$ to the flow head for predicting velocity, respectively:

$$\mathcal{L} = \alpha \mathcal{L}_{\text{NTP}} + \mathcal{L}_{\text{FM}}. \tag{4}$$

## 3.2 Training Recipe

Existing UMMs, such as Show-o [129], Janus-Pro [26], Transfusion [148], Chameleon [103], and Emu3 [115], are typically trained from LLMs, LMMs, or from scratch. These approaches aim to cultivate

Table 2: Trainable components and datasets in the training stages.

| | Trainable Components | Datasets | | |
|---|---|---|---|---|
| | | # Image-Text | # Video-Text | # Interleaved Data |
| **Stage-1** | Projector
Spatial (-Temporal) Fusion
Flow Head | 66M | WebVid [8]
Pandas [23] | OmniCorpus [60] |
| **Stage-2** | Full Model (w/o VAE) | 9M HQ Und.
16M HQ Gen. | OpenVid-1M [81] Gen.
1.5M Internal Data Gen.
1.6M Video Und. | VIST [47]
CoMM [24] |

visual generative modeling capabilities while preserving language modeling proficiency. However, this process often relies on web-scale, high-quality text corpora, which are prohibitively expensive to collect. Consequently, the lack of such resources can lead to a degradation in language knowledge and modeling performance. To address this challenge, we adopt a two-stage training recipe (as shown in Table 2) that effectively retains language knowledge while simultaneously developing visual generation capabilities, without requiring a massive text corpus.

**Stage-1.** Before the two-stage training, we have pre-distilled the semantic layers $\mathcal{S}(\cdot)$ (implementation details can be found in Section 4). The first stage only involves trainable components of the projector, spatial (-temporal) fusion, and flow head. In this stage, we train these components using autoregressive modeling and flow matching using around 66M image-text pairs and progressively add interleaved data and video-text pairs.

**Stage-2.** Subsequently, we tune the full model using 9M high-quality multimodal understanding instruction data, 16M high-quality visual generation data filtered from the 66M image-text pairs, and 1.6M video understanding data.

**Scaling Up.** After the training of the small-sized model with approximately 1.5B LLM parameters, we resume the pre-trained flow head for the larger model with 7B LLM parameters and introduce a lightweight MLP transformation to align the hidden size, allowing it to quickly adapt to the larger model and converge.

## 4 Experiments

Dataset curation and implementation details are provided in the Appendix A.

### 4.1 Multimodal Understanding on Images and Videos

**Quantitative Results.** Table 3 highlights the performance of our models on multimodal understanding benchmarks, evaluated across metrics such as MME [39], GQA [49], SEED-Bench [55], MM-Bench [76], MMU [138], MMStar [21], and AI2D [51]. As shown in the table, both the 1.5B and 7B variants of our model consistently outperform state-of-the-art models across many metrics. For models with similar parameter sizes (1.5B), our model achieves the best scores on MME-p and MMU-val benchmarks while delivering competitive performance on GQA and SEED-Bench metrics. When

Table 3: Evaluation on multimodal understanding benchmarks. # Params. indicates the number of parameters of base LLM. * indicates the method uses two distinct models or sets of parameters for multimodal understanding and generation, respectively. † indicates the Show-o2 models fine-tuned using video understanding data. Und. indicates "understanding". Results in gray indicate the performance of und. only models or models with total parameters more than 13B.

| Types | Models | # Params. | MME↑ (p) | GQA↑ | SEED↑ (all) | MMB↑ (en) | MMMU↑ (val) | MMStar↑ | AI2D↑ |
|---|---|---|---|---|---|---|---|---|---|
| Und. Only | LLaVA-v1.5 [72] | 7B | 1510.7 | 62.0 | 58.6 | 64.3 | - | - | - |
| | Qwen-VL-Chat [6] | 7B | 1487.6 | 57.5 | 58.2 | 60.6 | - | - | 57.7 |
| | LLaVA-OV [56] | 7B | 1580.0 | - | - | 80.8 | 48.8 | 57.5 | 81.4 |
| Unify via Assembling Tailored Models | NExT-GPT [129] | 13B | - | - | 57.5 | 58.0 | - | - | - |
| | SEED-X [40] | 17B | 1457.0 | 49.1 | 66.5 | 70.1 | 35.6 | - | - |
| | MetaMorph [107] | 8B | - | - | - | 71.8 | 75.2 | - | - |
| | TokenFlow-XL* [90] | 14B | 1551.1 | 62.5 | 72.6 | 76.8 | 43.2 | - | 75.9 |
| | ILLUME [112] | 7B | 1445.3 | - | 72.9 | 75.1 | 38.2 | - | 71.4 |
| Native Unified | BAGEL [32] | 14B | 1687.0 | - | - | 85.0 | 55.3 | - | - |
| | Show-o [129] | 1.3B | 1097.2 | 58.0 | 51.5 | - | 27.4 | - | - |
| | JanusFlow [80] | 1.5B | 1333.1 | 60.3 | 70.5 | 74.9 | 29.3 | - | - |
| | SynerGen-VL [58] | 2.4B | 1381.0 | - | - | 53.7 | 34.2 | - | - |
| | Janus-Pro [26] | 1.5B | 1444.0 | 59.3 | 68.3 | 75.5 | 36.3 | - | - |
| | **Show-o2 (Ours)** | 1.5B | 1450.9 | 60.0 | 65.6 | 67.4 | 37.1 | 43.4 | 69.0 |
| | Emu3 [115] | 8B | - | 60.3 | 68.2 | 58.5 | 31.6 | - | 70.0 |
| | VILA-U [124] | 7B | 1401.8 | 60.8 | 59.0 | - | - | - | - |
| | MUSE-VL [130] | 7B | - | - | 69.1 | 72.1 | 39.7 | 49.6 | 69.8 |
| | Liquid [119] | 8B | 1448.0 | 61.1 | - | - | - | - | - |
| | Janus-Pro [26] | 7B | 1567.1 | 62.0 | 72.1 | 79.2 | 41.0 | - | - |
| | Mogao [66] | 7B | 1592.0 | 60.9 | **74.6** | 75.0 | 44.2 | - | - |
| | **Show-o2 (Ours)** | 7B | **1620.5** | **63.1** | 69.8 | **79.3** | **48.9** | **56.6** | **78.6** |

Table 4: Evaluation on video understanding benchmarks. # Params. denotes the number of parameters in the base LLM, while # Frames represents the maximum number of video frames used during training and inference. Und. stands for understanding. † marks the Show-o2 models that have been fine-tuned on video understanding data. All results are reported in terms of zero-shot accuracy.

| Model | # Params. | # Frames | ActNet-QA | MVBench | NExT-QA | PerceptionTest | LongVideoBench | VideoMME |
|---|---|---|---|---|---|---|---|---|
| | | | test | test | mc | val | val | wo/w-subs |
| *Proprietary Und. Only Models* | | | | | | | | |
| GPT-4V [82] | - | - | 57.0 | 43.5 | - | - | 61.3 | 59.9/63.3 |
| GPT-4o [83] | - | - | - | - | - | - | 66.7 | 71.9/77.2 |
| Gemini-1.5-Flash [104] | - | - | 55.3 | - | - | - | 61.6 | 70.3/75.0 |
| Gemini-1.5-Pro [104] | - | - | 57.5 | - | - | - | 64.0 | 75.0/81.3 |
| *Open-source Und. Only Models* | | | | | | | | |
| VILA [70] | 40B | - | 58.0 | - | 67.9 | 54.0 | - | 60.1/61.1 |
| PLLaVA [132] | 34B | 16 / 16 | 60.9 | 58.1 | - | - | 53.2 | - |
| LongVA [144] | 7B | - | 50.0 | - | 68.3 | - | - | 52.6/54.3 |
| IXC-2.5 [143] | 7B | 64 / 64 | 52.8 | 69.1 | 71.0 | 34.4 | - | 55.8/58.8 |
| LLaVA-OV [56] | 7B | 32 / 32 | 56.6 | 56.7 | 79.4 | 57.1 | 56.5 | 58.2/61.5 |
| VideoLLaMA2 [30] | 7B | 16 / 16 | 50.2 | 54.6 | - | 51.4 | - | 47.9/50.3 |
| *Unified Multimodal Models* | | | | | | | | |
| **Show-o2†(Ours)** | 1.5B | 32 / 32 | 52.7 | 49.8 | 72.1 | 56.1 | 49.2 | 48.0/51.6 |
| **Show-o2†(Ours)** | 7B | 16 / 32 | 56.4 | 55.8 | 79.0 | 61.9 | 55.5 | 57.4/60.9 |

compared to larger models with approximately 7B parameters, our models surpass state-of-the-art models such as Janus-Pro and even the significantly larger TokenFlow-XL model (14B parameters) in metrics including MME-p, GQA, MMMU-val, MMStar, and AI2D, while maintaining competitive performance on SEED-Bench and MM-Bench. In addition, we present the video understanding performance of Show-o2† in Table 4.

**Qualitative Results.** Fig. 2 showcases the multimodal understanding capabilities of our model. As demonstrated, the model excels at answering general-purpose questions about an image. Specifically, it can provide detailed descriptions of an image, count objects, and recognize text within the image. Besides, the model can leverage its world knowledge to offer step-by-step instructions for preparing daily drinks like an avocado milkshake and supports bilingual question-answering, highlighting its versatility and practical utility.

Table 5: Evaluation on the GenEval [41] benchmark. Gen. denotes "generation". # Params. indicates the number of parameters of base LLM. # Data. indicates the number of image-text pairs used for visual generation during training. * means the method uses two distinct models for multimodal understanding and generation, respectively. Obj.: Object. Attri.: Attribute. Our results are obtained using rewritten prompts. + indicates the additional data required by the pretrained diffusion models.

| Type | Method | # Params. | # Data | Single Obj. | Two Obj. | Counting | Colors | Position | Color Attri. | Overall↑ |
|---|---|---|---|---|---|---|---|---|---|---|
| Gen. Only | SD3-Medium [37] | - | - | 0.99 | 0.94 | 0.72 | 0.89 | 0.33 | 0.60 | 0.74 |
| Unifying via Assembling Tailored Models | SEED-X [40] | 17B | 158M+ | 0.97 | 0.58 | 0.26 | 0.80 | 0.19 | 0.14 | 0.49 |
| | TokenFlow-XL* [90] | 14B | 60M | 0.95 | 0.60 | 0.41 | 0.81 | 0.16 | 0.24 | 0.55 |
| | ILLUME [112] | 7B | 15M+ | 0.99 | 0.86 | 0.45 | 0.71 | 0.39 | 0.28 | 0.61 |
| | MetaQuery-XL [84] | 7B | 28M+ | - | - | - | - | - | - | 0.80 |
| Native Unified | Show-o [129] | 1.3B | 2.0B | 0.98 | 0.80 | 0.66 | 0.84 | 0.31 | 0.50 | 0.68 |
| | Emu3 [115] | 8B | - | - | - | - | - | - | - | 0.66 |
| | MUSE-VL [130] | 7B | 24M | | | | | | | 0.57 |
| | Transfusion [148] | 7B | 3.5B | - | - | - | - | - | - | 0.63 |
| | D-DiT [64] | 2B | 40M | 0.97 | 0.80 | 0.54 | 0.76 | 0.32 | 0.50 | 0.65 |
| | Janus-Pro [26] | 7B | 144M | 0.99 | 0.89 | 0.59 | 0.90 | 0.79 | 0.66 | 0.80 |
| | BAGEL [32] | 14B | 1600M | 0.98 | 0.95 | 0.84 | 0.95 | 0.78 | 0.77 | 0.88 |
| | Mogao [66] | 7B | - | 1.00 | 0.97 | 0.83 | 0.93 | 0.84 | 0.80 | **0.89** |
| | **Show-o2 (Ours)** | 1.5B | 66M | 0.99 | 0.86 | 0.55 | 0.86 | 0.46 | 0.63 | 0.73 |
| | **Show-o2 (Ours)** | 7B | 66M | 1.00 | 0.87 | 0.58 | 0.92 | 0.52 | 0.62 | 0.76 |

Table 6: Evaluation on the DPG-Bench [45] benchmark. Gen. denotes "generation". # Params. indicates the number of parameters of base LLM. # Data. indicates the number of image-text pairs used for visual generation during training.

| Type | Method | # Params. | # Data | Global | Entity | Attribute | Relation | Other | Overall↑ |
|---|---|---|---|---|---|---|---|---|---|
| Gen. Only | Hunyuan-DiT [65] | 1.5B | - | 84.59 | 80.59 | 88.01 | 74.36 | 86.41 | 78.87 |
| | Playground v2.5 [57] | - | - | 83.06 | 82.59 | 81.20 | 84.08 | 83.50 | 75.47 |
| | PixArt-Σ [17] | - | - | 86.89 | 82.89 | 88.94 | 86.59 | 87.68 | 80.54 |
| | DALL-E 3 [10] | - | - | 90.97 | 89.61 | 88.39 | 90.58 | 89.83 | 83.50 |
| | SD3-Medium [37] | 2B | - | 87.90 | 91.01 | 88.83 | 80.70 | 88.68 | 84.08 |
| Native Unified | Emu3-DPO [115] | 8B | - | - | - | - | - | - | 81.60 |
| | Janus-Pro [26] | 7B | 144M | 86.90 | 88.90 | 89.40 | 89.32 | 89.48 | 84.19 |
| | Mogao [66] | 7B | - | 82.37 | 90.03 | 88.26 | 93.18 | 85.40 | 84.33 |
| | **Show-o2 (Ours)** | 1.5B | 66M | 87.53 | 90.38 | 91.34 | 90.30 | 91.21 | 85.02 |
| | **Show-o2 (Ours)** | 7B | 66M | 89.00 | 91.78 | 89.96 | 91.81 | 91.64 | **86.14** |

Table 7: Overall quantitative comparison of different methods on OneIG-Bench. Gen. denotes "generation". # Params. indicates the number of parameters of base LLM. # Data. indicates the number of image-text pairs used for visual generation during training.

| Type | Method | # Params. | # Data | Alignment↑ | Text↑ | Reasoning↑ | Style↑ | Diversity↑ |
|---|---|---|---|---|---|---|---|---|
| Gen. Only | SD3.5-Large [37] | 8B | - | 0.809 | 0.629 | 0.294 | 0.353 | 0.225 |
| | Flux.1-dev [54] | 12B | - | 0.786 | 0.523 | 0.253 | 0.368 | 0.238 |
| | SANA-1.5 (PAG) [127] | 4.8B | - | 0.765 | 0.069 | 0.217 | 0.401 | 0.216 |
| | Lumina-Image 2.0 [89] | 2.6B | 110M | 0.819 | 0.106 | 0.270 | 0.354 | 0.216 |
| | HiDream-I1-Full [44] | 17B | - | 0.829 | 0.707 | 0.317 | 0.347 | 0.186 |
| Unified Models | Show-o-512 [129] | 1.3B | 2B | 0.702 | 0.002 | 0.213 | 0.361 | 0.241 |
| | Janus-Pro [27] | 7B | 144M | 0.553 | 0.001 | 0.139 | 0.276 | 0.365 |
| | BLIP3-o [18] | 8B | 55M | 0.711 | 0.013 | 0.223 | 0.361 | 0.229 |
| | BAGEL [32] | 14B | 1600M | 0.769 | 0.244 | 0.173 | 0.367 | 0.251 |
| | OmniGen2 [118] | 7B | 150M | 0.804 | 0.680 | 0.271 | 0.377 | 0.242 |
| | **Show-o2 (Ours)** | 1.5B | 66M | 0.798 | 0.002 | 0.219 | 0.317 | 0.186 |
| | **Show-o2-1024×1024 (Ours)** | 1.5B | 66M | 0.798 | 0.125 | 0.274 | 0.351 | 0.186 |
| | **Show-o2 (Ours)** | 7B | 66M | 0.817 | 0.002 | 0.226 | 0.317 | 0.177 |

## 4.2 Visual Generation

**Image Generation.** We compare our model with the state-of-the-art approaches on GenEval [41], DPG-Bench [45], and OneIG [13] benchmarks in Tables 5, 6, and 7. One can observe that our model surpasses most approaches, including TokenFlow-XL, Show-o, Emu3, and Transfusion, on the GenEval benchmark. Compared to Janus-Pro, which was trained on a significantly larger dataset of 144M image-text pairs, our model achieves promising results with only 66M image-text pairs. On DPG-Bench evaluation, our model has demonstrated the best overall score compared to generation-only models such as SD3-Medium and unified models, including Emu3-DPO and Janus-Pro. On OneIG-Bench, our models also achieve competitive performance. We also show qualitative results in Fig. 2 to illustrate that our model can generate high-quality and realistic images.

**Video Generation.** We compare our model with the text-to-video and image-to-video generation models in Tables 8 and 9. One can observe that with only 2B parameters, our model outperforms models such as Show-1, Emu3, and VILA-U with more than 6B parameters. Besides, our model has demonstrated competitive performance compared to CogVideoX and Step-Video-T2V. We also provide qualitative results of the text-to-video and image-to-video generation capability of our model in the middle of Fig. 2.

*Multimodal Understanding*

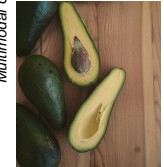

*Describe the image in detail.*

The image captures a collection of objects arranged on a white lace fabric. Dominating the scene is a beige scarf, its edges adorned with fringe. The scarf is not just a piece of clothing, but a canvas for the word "inspire", written in a black cursive font. Resting on the scarf is a black choker, its gold buckle adding a touch of elegance. The choker's position suggests it's ready to be worn, perhaps as a symbol of inspiration.

*How many avocados are in this image? How to make an avocado milkshake?*

There are five avocados in this image. To make an avocado milkshake, you'll need: 1. 2 ripe avocados. 2. 1 cup of milk (dairy or non-dairy). 3. 1/2 cup of ice cream. 4. 1/2 teaspoon of vanilla extract. 5. A blender.
Instructions: 1. Peel the avocados and remove the pit. 2. Add the avocados, milk, ice cream, and vanilla extract to the blender. 3. Blend on high speed until smooth and creamy. 4. Taste and adjust the consistency or sweetness as needed. 5. Pour into glasses and enjoy your delicious avocado milkshake!

The moon tells the sky
The sky tells the sea
The sea tells the tide
And the tide tells me

*告诉我图片中写着什么？*

图片上是一首诗，内容如下：The moon tells the sky. The sky tells the sea. The sea tells the tide. And the tide tells me.

*Text-to-Image Generation*

Be Kind

SHOW 02

*Text-to-Video Generation*

A curly-haired blonde girl smiles softly, her bright eyes joyful. She wears a pink lace dress, with rosy cheeks against a flowered background.

A breathtaking coastal beach in spring, where gentle waves caress the golden sand.

*Image-to-Video Generation*

Input image

*Mixed-Modality Generation*

The cemetery had green grass [BOI]

[EOI]and large trees.[BOI]

[EOI]The church was red[BOI]

[EOI]and had a pretty ceiling.[BOI]

My wife surprised me with a private helicopter ride to explore the wilderness.[BOI]

[EOI]We started looking out into the mountains[BOI]

[EOI]then went into them looking at a lake.[BOI]

[EOI]It was so beautiful seeing the mountains.[BOI]

Figure 2: Multimodal understanding and generation examples.

Table 8: Comparison with text-to-video models on the VBench [48] benchmark. # Params. indicates the number of total parameters for video generation including the base LLM and flow head. QS: Quality Score, SS: Semantic Score, SC: Subject Consistency, BC: Background Consistency, TF: Temporal Flickering, MS: Motion Smoothness, DD: Dynamic Degree, AQ: Aesthetic Quality, IQ: Imaging Quality, OC: Object Class, MO: Multiple Objects, HA: Human Action, C: Color, SR: Spatial Relationship, S: Scene, AS: Appearance style, TS: Temporal Style, OC': Overall Consistency.

| Models | # Params. | Total | QS | SS | SC | BC | TF | MS | DD | AQ | IQ | OC | MO | HA | C | SR | S | AS | TS | OC' |
|---|---|---|---|---|---|---|---|---|---|---|---|---|---|---|---|---|---|---|---|---|
| ModelScope [114] | 1.7B | 75.75 | 78.05 | 66.54 | 89.87 | 95.29 | 98.28 | 95.79 | 66.39 | 52.06 | 58.57 | 82.25 | 38.98 | 92.40 | 81.72 | 33.68 | 39.26 | 23.39 | 25.37 | 25.67 |
| LaVie [116] | 3B | 77.08 | 78.78 | 70.31 | 91.41 | 97.47 | 98.30 | 96.38 | 49.72 | 54.94 | 61.90 | 91.82 | 33.32 | 96.80 | 86.39 | 34.09 | 52.69 | 23.56 | 25.93 | 26.41 |
| OpenSoraPlan V1.3 [67] | - | 77.23 | 80.14 | 65.62 | 97.79 | 97.24 | 99.20 | 99.05 | 30.28 | 60.42 | 56.21 | 85.56 | 43.58 | 86.80 | 79.30 | 51.61 | 36.73 | 20.03 | 22.47 | 24.47 |
| Show-1 [141] | 6B | 78.93 | 80.42 | 72.98 | 95.53 | 98.02 | 99.12 | 98.24 | 44.44 | 57.35 | 58.66 | 93.07 | 45.47 | 95.60 | 86.35 | 53.50 | 47.03 | 23.06 | 25.28 | 27.46 |
| AnimateDiff-V2 [43] | - | 80.27 | 82.90 | 69.75 | 95.30 | 97.68 | 98.75 | 97.76 | 40.83 | 67.16 | 70.10 | 90.90 | 36.88 | 92.60 | 87.47 | 34.60 | 50.19 | 22.42 | 26.03 | 27.04 |
| Gen-2 [1] | - | 80.58 | 82.47 | 73.03 | 97.61 | 97.61 | 99.56 | 99.58 | 18.89 | 66.96 | 67.42 | 90.92 | 55.47 | 89.20 | 89.49 | 66.91 | 48.91 | 19.34 | 24.12 | 26.17 |
| Pika-1.0 [2] | - | 80.69 | 82.92 | 71.77 | 96.94 | 97.36 | 99.74 | 99.50 | 47.50 | 62.04 | 61.87 | 88.72 | 43.08 | 86.20 | 90.57 | 61.03 | 49.83 | 22.26 | 24.22 | 25.94 |
| VideoCrafter-2.0 [16] | - | 80.44 | 82.20 | 73.42 | 96.85 | 98.22 | 98.41 | 97.73 | 42.50 | 63.13 | 67.22 | 92.55 | 40.66 | 95.00 | 92.92 | 35.86 | 55.29 | 25.13 | 25.84 | 28.23 |
| CogVideoX [137] | 5B | 81.61 | 82.75 | 77.04 | 96.23 | 96.52 | 98.66 | 96.92 | 70.97 | 61.98 | 62.90 | 85.23 | 62.11 | 99.40 | 82.81 | 66.35 | 53.20 | 24.91 | 25.38 | 27.59 |
| Kling [4] | - | 81.85 | 83.39 | 75.68 | 98.33 | 97.60 | 99.30 | 99.40 | 46.94 | 61.21 | 65.62 | 87.24 | 68.05 | 93.40 | 89.90 | 73.03 | 50.86 | 19.62 | 24.17 | 26.42 |
| Step-Video-T2V [79] | 30B | 81.83 | 84.46 | 71.28 | 98.05 | 97.67 | 99.40 | 99.08 | 53.06 | 61.23 | 70.63 | 80.56 | 50.55 | 94.00 | 88.25 | 71.47 | 24.38 | 23.17 | 26.01 | 27.12 |
| Gen-3 [3] | - | 82.32 | 84.11 | 75.17 | 97.10 | 96.62 | 98.61 | 99.23 | 60.14 | 63.34 | 66.82 | 87.81 | 53.64 | 96.40 | 80.90 | 65.09 | 54.57 | 24.31 | 24.71 | 26.69 |
| Emu3 [115] | 8B | 80.96 | - | - | 95.32 | 97.69 | - | 98.93 | 79.27 | 59.64 | - | 86.17 | 44.64 | 77.71 | - | 68.73 | 37.11 | 20.92 | - | - |
| VILA-U [124] | 7B | 74.01 | 76.26 | 65.04 | - | - | - | - | - | - | - | - | - | - | - | - | - | - | - | - |
| HaploOmni [125] | 9B | 78.10 | - | - | 96.40 | 97.60 | - | 96.80 | 65.30 | - | - | - | - | - | - | - | 34.60 | - | - | - |
| **Show-o2 (Ours)** | 2B | 81.34 | 82.10 | 78.31 | 97.28 | 96.78 | 97.68 | 98.25 | 40.83 | 65.15 | 67.06 | 94.81 | 76.01 | 95.20 | 80.89 | 62.61 | 57.67 | 23.29 | 25.27 | 27.00 |

Table 9: Comparison with image-to-video models on the VBench [48] benchmark.

| Models | I2V Subject | I2V Background | Camera Motion | Subject Consistency | Background Consistency | Temporal Flickering | Motion Smoothness | Dynamic Degree | Aesthetic Quality | Imaging Quality |
|---|---|---|---|---|---|---|---|---|---|---|
| DynamiCrafter-1024 [131] | 96.71 | 96.05 | 35.44 | 95.69 | 97.38 | 97.63 | 97.38 | 47.40 | 66.46 | 69.34 |
| SEINE-512x320 [25] | 94.85 | 94.02 | 23.36 | 94.20 | 97.26 | 96.72 | 96.68 | 34.31 | 58.42 | 70.97 |
| I2VGen-XL [145] | 96.74 | 95.44 | 13.32 | 96.36 | 97.93 | 98.48 | 98.31 | 24.96 | 65.33 | 69.85 |
| Animate-Anything [31] | 98.54 | 96.88 | 12.56 | 98.90 | 98.19 | 98.14 | 98.61 | 2.68 | 67.12 | 72.09 |
| ConsistI2V [92] | 94.69 | 94.57 | 33.60 | 95.27 | 98.28 | 97.56 | 97.38 | 18.62 | 59.00 | 66.92 |
| VideoCrafter-I2V [15] | 90.97 | 90.51 | 33.58 | 97.86 | 98.79 | 98.19 | 98.00 | 22.60 | 60.78 | 71.68 |
| SVD-XT-1.1 [11] | 97.51 | 97.62 | - | 95.42 | 96.77 | 99.17 | 98.12 | 43.17 | 60.23 | 70.23 |
| MarDini [74] | 98.78 | 96.46 | - | - | - | - | - | - | - | - |
| **Show-o2 (Ours)** | 96.94 | 98.83 | 28.41 | 93.83 | 97.45 | - | 97.76 | 25.85 | 61.92 | 69.87 |

## 4.3 Mixed-Modality Generation

We demonstrate mixed-modality generation capabilities of our model using downstream task visual storytelling dataset [47] in Fig. 2. During fine-tuning, given an interleaved image-text sequence, we apply noise to all images in the sequence with a probability of 0.3. Otherwise, we randomly retain a number of the earlier images in the sequence and only apply noise to the later ones. Benefiting from the general interleaved sequence format mentioned in 3.1, our model can predict the [BOI] once it begins to generate an image. Upon detecting the [BOI] token, noises will be appended to the sequence to gradually generate an image. The generated text tokens and images will be served as context to continue generating the following output. Fig. 2 includes two examples demonstrating our model's ability to interleavely generate coherent text and images, vividly narrating a story.

## 4.4 Ablation Studies

We show the pilot study results in Table 10, which validated the effect of spatial (-temporal) fusion on multimodal understanding and generation performance. For efficiency, we adopt LLaMA-3.2-1B as the base language model and use only around 1M multimodal understanding data and the ImageNet-1K generation data [33]. Under the same training settings, there are improvements in terms of both multimodal understanding and generation metrics, including MME-p, GQA, and FID-5K. This validates that the involved semantic and low-level features in the fusion mechanism would potentially help both the multimodal generation and understanding capabilities to some extent.

Table 10: Impact of spatial (-temporal) fusion.

| | MME$-$p $\uparrow$ | GQA $\uparrow$ | POPE $\uparrow$ | FID-5K $\downarrow$ |
|---|---|---|---|---|
| w/o Fusion | 1164.7 | 56.2 | 82.6 | 21.8 |
| w Fusion | **1187.8** | **57.6** | 82.6 | **20.5** |

Table 11 provides the effect of training stages on the generation performance on the GenEval and DPG-Bench benchmarks. One can observe that stage-2 training consistently and significantly improves both metrics, which validates the importance of the second stage.

Table 11: Effect of training stages.

| Stage-1 | Stage-2 | GenEval | DPG-Bench |
|---|---|---|---|
| ✓ | | 0.63 | 83.28 |
| ✓ | ✓ | **0.73** | **84.70** |

We perform ablation studies to examine the effect of classifier-free guidance (CFG) and inference steps on the performance using the 1.5B model. As shown in Table 12, increasing the CFG guidance scale and inference steps (in a range) would potentially improve the GenEval and DPG-Bench scores. However, the improvements of the GenEval score are not significant when the CFG guidance is set as larger than 5.0.

Table 12: Effect of CFG guidance and inference steps.

| CFG guidance | Inference steps | GenEval | DPG-Bench |
|---|---|---|---|
| 2.5 | 50 | 0.65 | 81.6 |
| 5.0 | 50 | 0.71 | 83.9 |
| 7.5 | 50 | 0.71 | 84.8 |
| 10 | 50 | 0.71 | **85.0** |
| 7.5 | 25 | 0.71 | 84.6 |
| 7.5 | 100 | **0.73** | 84.7 |

Table 13: Impact of training recipe on text-only performance. One-stage training denotes full-parameter-co-training on image-text pairs and the text-only RefinedWeb [88] data. Note that the curated multimodal understanding data consists of text-only instructional data. We perform the evaluation under the same setting using the lm-evaluation-harness tool.

| Models | # Params. | Training Recipe | MMLU | GPQA | GSM8K | HumanEval |
|---|---|---|---|---|---|---|
| Qwen2.5 Instruct [134] | 1.5B | - | $60.20 \pm 0.39$ | $28.12 \pm 2.13$ | $51.86 \pm 1.38$ | $35.37 \pm 3.74$ |
| Show-o2 (Ours) | 1.5B | One-stage training with RefinedWeb | $28.25 \pm 0.38$ | $25.00 \pm 2.05$ | $4.55 \pm 0.57$ | $3.05 \pm 1.35$ |
| Show-o2 (Ours) | 1.5B | Our two-stage training | $56.75 \pm 1.37$ | $29.24 \pm 2.15$ | $49.43 \pm 1.38$ | $35.54 \pm 3.70$ |
| Qwen2.5 Instruct [134] | 7B | - | $71.75 \pm 0.36$ | $32.37 \pm 2.21$ | $82.49 \pm 1.05$ | $65.24 \pm 3.73$ |
| Show-o2 (Ours) | 7B | One-stage training with RefinedWeb | $28.43 \pm 0.21$ | $26.34 \pm 2.08$ | $1.52 \pm 0.34$ | $4.01 \pm 1.25$ |
| Show-o2 (Ours) | 7B | Our two-stage training | $70.73 \pm 0.36$ | $31.47 \pm 2.22$ | $75.28 \pm 1.19$ | $70.73 \pm 3.56$ |

Table 13 shows that our models effectively preserve language knowledge and achieve performance comparable to the original Qwen2.5-1.5B and Qwen2.5-7B Instruct models. In contrast, direct one-stage full-parameter-co-training with textual data such as RefinedWeb results in substantial performance degradation, highlighting the necessity of the two-stage training approach when high-quality corpora are unavailable.

As shown in Table 14, our ablation study reveals that increasing the number of image tokens significantly boosts performance across all tasks, even though the

Table 14: Impact of image token count on chart, text, and document VQA.

| Models | # Params. | # Image tokens | ChartQA | DocVQA$_{val}$ | InfoVQA$_{val}$ | TextVQA$_{val}$ |
|---|---|---|---|---|---|---|
| LLaVA-OV | 7B | 729 | 56.24 | 62.71 | 39.59 | 66.19 |
| **Show-o2** | 7B | 729 | 48.00 | 59.34 | 42.31 | 62.92 |
| **Show-o2** | 7B | $5 \times 729$ | 66.92 | 77.26 | 45.80 | 71.54 |

model was trained with a fixed image resolution. Using the AnyRes strategy at inference time consistently improves results, highlighting the benefit of higher token counts for capturing fine-grained details. When compared to the baseline LLaVA-OV-7B, our model achieves comparable results on DocVQA, InfoVQA, and TextVQA validation sets, but underperforms on ChartQA. We attribute this gap to the limited chart-related data available during semantic layer distillation, which constrains the model's ability to capture chart-specific information. We believe that incorporating more OCR and document-centric data into the distillation process will further strengthen the unified model's OCR and document understanding capabilities.

## 5 Conclusion

This paper proposed native unified multimodal models, *i.e.,* Show-o2, scalable for multimodal understanding and generation, image and video modalities, by integrating 3D causal VAE, autoregressive modeling, and flow matching. A dual-path of spatial (-temporal) fusion mechanism guided the construction of unified visual representations with both high- and low-level features. A two-stage training recipe enables effective learning of unified capabilties, resulting in a versatile model capable of handling diverse tasks, including multimodal understanding and image/video generation. Extensive experiments demonstrate the model's state-of-the-art performance across various benchmarks.

**Ethics Statement.** We use CC12M [14], COYO [12], LAION-Aesthetic-12M, AI-synthetic data, and internally collected video data, all of which are strictly filtered to exclude copyrighted materials and visible watermarks during data curation. A safety checker is integrated into our inference pipeline to mitigate potential misuse. We have conducted repeated evaluations and observed stable results with minimal variations. However, our experimental comparisons follow prior literature that did not report error bars. For language-only evaluations, we report the standard deviations.

**Acknowledgments.** This research is supported by the National Research Foundation, Singapore under its AI Singapore Programme (AISG Award No: AISG3-RP-2022-030).

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

# A    Technical Appendices and Supplementary Material

## A.1    Experimental Setup

**Datasets.** The curated approximately 66M image-text pairs consist of images with a resolution of at least 512 pixels in width and height. The images are filtered from CC12M [14], COYO [12], LAION-Aesthetic-12M* and AI synthetic data. The images are recaptioned by LMMs except for the synthetic data. The 9M high-quality multimodal understanding instruction data is curated from Densefusion-1M [63], and LLaVA-OneVision [56].

**Implementation Details.** The semantic layers $\mathcal{S}(\cdot)$ are pre-distilled from SigLIP-so400m-patch14-384* over 200K iterations, using a batch size of 512 and a cosine-scheduled learning rate of 2e-5. During distillation, Eq. 1 is applied to the visual latents with only a probability of 0.3 in the last 20K iterations. The input image resolution of 3D causal VAE encoder with $2 \times 2$ patch embedding layer is set as $432 \times 432$ to get $729 = 27 \times 27$ visual latents, which matches the ones extracted by SigLIP. Once distilled, the semantic layers $\mathcal{S}(\cdot)$ are capable of extracting rich semantic features from both clean and noised visual latents. In statistics, the extracted features from clean visual latents by $\mathcal{S}(\cdot)$ have converged to an average cosine similarity of around 0.9 with those extracted by the original SigLIP on the curated 66M image-text pairs. We interpolate the position embeddings in the bicubic mode when involving other image/video resolutions.

Our models build upon two LLM variants, *i.e.*, Qwen2.5-1.5B-Instruct [134] and Qwen2.5-7B-Instruct [134], respectively. We adopt 3D causal VAE proposed in Wan2.1 [109] with $8\times$ and $4\times$ spatial and temporal compression, respectively. In stage 1, we first train the 1.5B variant for 150K iterations using AdamW optimizer with a constant learning rate of 0.0001 on the curated 66M image-text pairs in a resolution of $432 \times 432$. The context length of single image-text pairs is set as 1024. The total batch sizes for multimodal understanding and generation are 128 and 384, respectively. $\alpha$ in Eq. 4 is set as 0.2. For visual generation data, the caption is dropped with a probability of 0.1 to enable the classifier-free guidance. This training process roughly takes one and a half days using 64 H100 GPUs. Subsequently, we replace the generation data with 16M high-quality data (filtered from 66M image-text pairs) and continue to train for 40K iterations. In stage 2, we follow the training strategies in LLaVA-OneVision [56] to train the 1.5B model using around 9M multimodal instructional and 16M high-quality generation data for a total of around 35K iterations. $\alpha$ in Eq. 4 is set as 1.0. The stage 2 training process takes around 15 hours. For models with mixed-modality and video generation capabilities, we progressively add video-text and interleaved data in stage 1. For video data, we randomly sample a 2s 480p or $432\times432$ clips with 17 frames from each video with an interval of 3 frames. The context length at this time is set as 7006. In stage 2, high-quality video-text and interleaved data are added to further improve video and mixed-modality generation capabilities.

To futher improve the image generation and text rendering quality, we further train the small-scale model on images with higher resoluton ($512 \times 512$ and $1024 \times 1024$) and involve an additional text-rich image data, *i.e.*, a subset of TextAtlas [110].

Building on the pre-trained image-level Show-o2 models, we enhance their video understanding capabilities by further fine-tuning on 1.6M video samples from [146], together with 1.1M image-level samples from the earlier stage. We adopt the same video training and inference settings as LLaVA-OneVision. The evaluation results are shown in Table 4.

In the training of our model based on the 7B LLM variant, we resume the flow head pre-trained based on the 1.5B model and additionally train the newly initialized spatial (-temporal) fusion, projector, and MLP transformations for 3K iterations with 2K warm-up steps to align the hidden size and then further train spatial (-temporal) fusion, the projector, MLP transformations, and the flow head together. Following that, we conduct the training stages 1 and 2 in the same manner as those of the 1.5B model. The whole training process of our 7B model takes approximately 2 and a half days using 128 H100 GPUs. We do not include interleaved and video data in the training stages of the larger model due to the huge computational cost and training duration.

---

*  https://huggingface.co/datasets/dclure/laion-aesthetics-12m-umap
*  https://huggingface.co/google/siglip-so400m-patch14-384

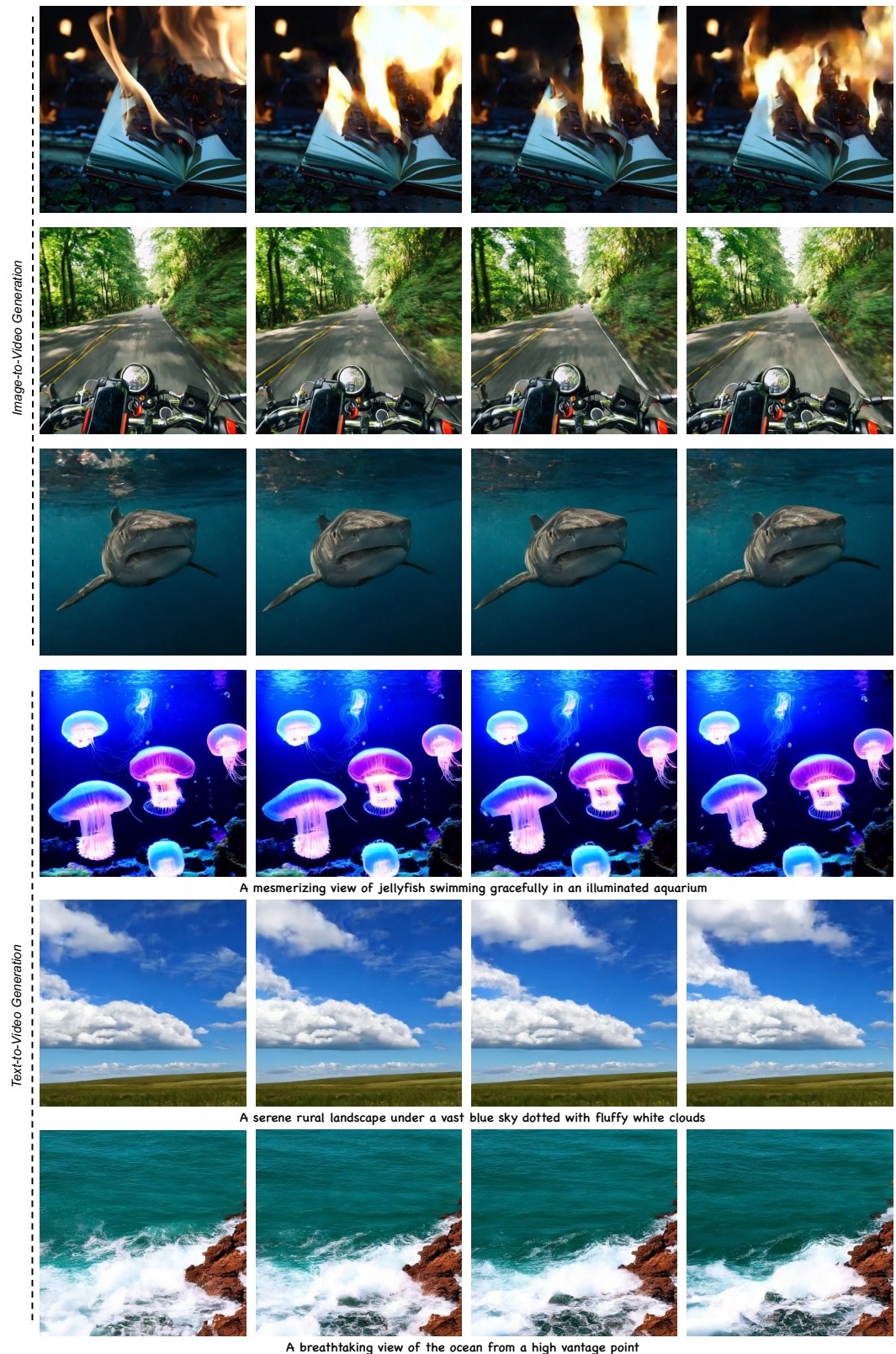

Figure 3: Text-to-video and image-to-video generation examples.

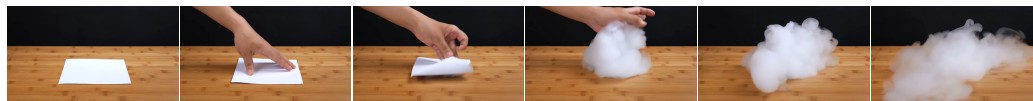

*What is the unusual aspect in this video?*

The unusual aspect in this video is the formation of a large amount of white smoke from a small paper, creating a dramatic and visually captivating cloud.

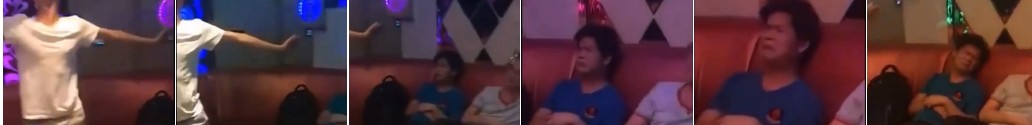

*What's funny about this video?*

LLaVA-OV-7B:
The video is funny because the person in the white shirt and gray shorts is seen dancing energetically, while the person in the blue shirt remains mostly motionless, creating a humorous contrast.

Show-o2-7B (Ours):
The funny part is that the person in the white shirt is singing or speaking into the microphone, while everyone else is sitting on the couch and looking at the person, possibly in amusement or confusion.

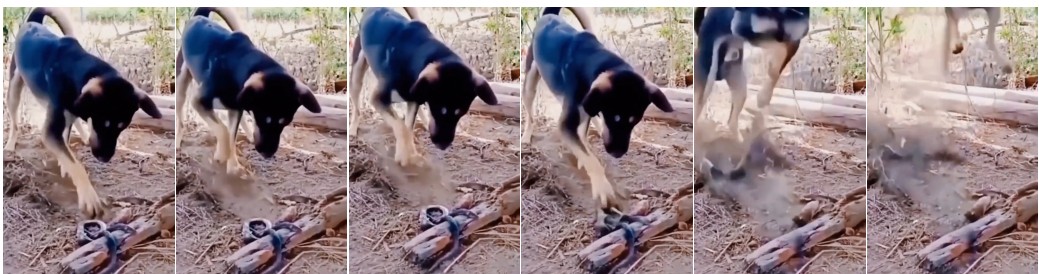

*What's funny about this video?*

LLaVA-OV-7B:
The funny part is the dog's reaction to the snake, which it tries to catch and then playfully interacts with.

Show-o2-7B (Ours):
The dog's exaggerated reaction to the snakes makes the scene comical.

Figure 4: Video understanding examples.

## A.2 More Qualitative Results

Figures 3 and 4 privide more visualization results of video generation and understanding.

## A.3 Text Prompts

We provide the text prompts for image generation used in Fig. 2 below:

"Hyper-detailed image of a mature man with short, graying hair and deep blue eyes. He has a rugged, weathered face with a strong jawline and a slight beard. His expression is thoughtful and introspective. The lighting is dramatic, highlighting the contours of his face. The photo is in 8K resolution, capturing every wrinkle and pore. "

"A soft, natural portrait photograph captures a young woman with fair skin and long, ash-blonde hair cascading gently over her shoulders, her striking light blue eyes subtly enhanced with natural makeup and a gentle, calm smile playing on her lips. She wears a cozy, cream-colored winter sweater and a delicate woolen scarf adorned with subtle snowflake patterns, positioned slightly off-center, creating a sense of relaxed elegance. Behind her, a softly blurred snowy Moscow street scene unfolds, with traditional architecture and the diffused, golden glow of a winter afternoon contributing to a serene and contemplative atmosphere. At the very bottom of the frame, in simple, elegant lettering, appears the phrase "BE KIND". "

"A vibrant, highly detailed close-up of a colorful parrot perched on a branch, featuring intricate feather textures, vivid colors (red, blue, green, yellow), and a tropical rainforest background. The parrot's

Table 15: Comparative analysis of selected unified multimodal models based on the utilization of visual representations and type of unified modeling for multimodal understanding and generation. In this context, **native und. & gen.** refers to the direct decoding of output sequences into texts, images, and videos, as opposed to serving as conditions for decoding using external pre-trained decoders like Stable Diffusion. * indicates the method uses two distinct models for multimodal understanding and generation, respectively.

| Methods | Und. & Gen. Representation | | | Type of Unified Modeling | | |
| --- | --- | --- | --- | --- | --- | --- |
| | Unified | Decoupled | Support Video | Native Und. & Gen. | Assembling Tailored Models | Paradigm |
| Chameleon [103] | ✓ | | ✗ | ✓ | | AR |
| Show-o [129] | ✓ | | ✗ | ✓ | | AR + Diff. |
| Transfusion [148] | ✓ | | ✗ | ✓ | | AR + Diff. |
| VILA-U [124] | ✓ | | ✓ | ✓ | | AR |
| Emu3 [115] | ✓ | | ✓ | ✓ | | AR |
| MonoFormer [147] | ✓ | | ✗ | ✓ | | AR + Diff. |
| Dual-Diffusion [64] | ✓ | | ✗ | ✓ | | Diff. |
| SynerGen-VL [58] | ✓ | | ✗ | ✓ | | AR |
| MMAR [135] | ✓ | | ✗ | ✓ | | AR + MAR |
| MUSE-VL [130] | ✓ | | ✗ | ✓ | | AR |
| Orthus [53] | ✓ | | ✗ | ✓ | | AR + Diff. |
| Liquid [119] | ✓ | | ✗ | ✓ | | AR |
| LlamaFusion [96] | ✓ | | ✗ | ✓ | | AR + Diff. |
| UGen [100] | ✓ | | ✗ | ✓ | | AR |
| UniDisc [99] | ✓ | | ✗ | ✓ | | Diff. |
| UniToken [50] | ✓ | | ✗ | ✓ | | AR |
| Harmon [123] | ✓ | | ✗ | ✓ | | AR+MAR |
| DualToken [97] | ✓ | | ✗ | ✓ | | AR |
| UniTok [78] | ✓ | | ✗ | ✓ | | AR |
| Selftok [111] | ✓ | | ✗ | ✓ | | AR |
| Muddit [95] | ✓ | | ✗ | ✓ | | Diff. |
| MMaDA [136] | ✓ | | ✗ | ✓ | | Diff. |
| HaploOmni [125] | ✓ | | ✓ | ✓ | | AR + Diff. |
| TokLIP [69] | ✓ | | ✗ | ✓ | | AR |
| Show-o2 (Ours) | ✓ | | ✓ | ✓ | | AR + Diff. |
| Janus-Series [26, 27, 80] | | ✓ | ✗ | ✓ | | AR (+Diff.) |
| VARGPT [149] | | ✓ | ✗ | ✓ | | AR |
| UnidFluid [38] | | ✓ | ✗ | ✓ | | AR + MAR |
| OmniMamba [150] | | ✓ | ✗ | ✓ | | AR |
| Mogao [66] | | ✓ | ✗ | ✓ | | AR + Diff. |
| BAGEL [32] | | ✓ | ✓ | ✓ | | AR + Diff. |
| Fudoki [113] | | ✓ | ✗ | ✓ | | Diff. |
| UniGen [105] | | ✓ | ✗ | ✓ | | AR + Diff. |
| NExT-GPT [121] | | ✓ | ✓ | | ✓ | AR + Diff. |
| CoDI [102] | | ✓ | ✓ | | ✓ | AR + Diff. |
| DreamLLM [36] | | ✓ | ✗ | | ✓ | AR + Diff. |
| SEED-X [40] | | ✓ | ✗ | | ✓ | AR + Diff. |
| MIO [117] | | ✓ | ✓ | | ✓ | AR + Diff. |
| CoDI-2 [101] | | ✓ | ✓ | | ✓ | AR + Diff. |
| MetaMorph [107] | | ✓ | ✗ | | ✓ | AR + Diff. |
| ILLUME [112] | | ✓ | ✗ | | ✓ | AR + Diff. |
| ILLUME+ [46] | | ✓ | ✗ | | ✓ | AR + Diff. |
| MetaQueries [84] | | ✓ | ✗ | | ✓ | AR + Diff. |
| Nexus-Gen [142] | | ✓ | ✗ | | ✓ | AR + Diff. |
| Ming-Lite-Uni [42] | | ✓ | ✗ | | ✓ | AR + Diff. |
| BLIP3-o [18] | | ✓ | ✗ | | ✓ | AR + Diff. |
| OpenUni [122] | | ✓ | ✗ | | ✓ | AR + Diff. |
| UniWorld [68] | | ✓ | ✗ | | ✓ | AR + Diff. |
| Ming-Omni [5] | | ✓ | ✓ | | ✓ | AR + Diff. |
| Pisces [133] | | ✓ | ✗ | | ✓ | AR + Diff. |
| TokenFlow* [90] | ✓ | | ✗ | | ✓ | AR |
| SemHiTok* [28] | ✓ | | ✗ | | ✓ | AR |

eyes are sharp and expressive, with a natural glint of light. The image is photorealistic, ultra HD (8K resolution), with soft natural lighting and a shallow depth of field, creating a blurred bokeh effect in the background. The scene is peaceful and lush, showcasing the beauty of nature. "

"A dark, moody room with a glowing neon sign on the wall that spells out 'SHOW O2' in bold, vibrant pink and blue colors. The neon light reflects softly on the polished concrete floor, creating a futuristic and artistic vibe. "

## A.4 Limitations and Broader Impacts

We found that our model is not good at rendering text on the image. We investigated our generation datasets and observed that the proportion of images with rendered texts is relatively small, which

potentially leads to bad text rendering. In addition, the generated images will lack details of the small objects because of the limited image resolution. To address this limitation, as outlined in the implementation details, we have enhanced the model by training it on higher-resolution data and incorporating image datasets rich in textual information.

Our models possess the ability to generate text and images, which may carry the risk of unintended misuse. As discussed in our Ethics Statement, we have conducted data filtering to avoid this issue and integrated a safety checker into our inference pipeline to mitigate potential misuse.

