# OpenReview forum: "Show-o2: Improved Native Unified Multimodal Models"
_NeurIPS.cc/2025/Conference — NeurIPS 2025 poster_

### Official Review · Reviewer_sCtb · 2025-06-28

**Clarity:** 3
**Significance:** 3
**Originality:** 3
**Rating:** 5
**Confidence:** 4

**Summary:**

In this paper, authors propose a unified multi-modal training mechanism, including model design, training stages and training data. They propose to apply next-token generation loss for text modality and flow matching for image generation. For image understanding and generation, they use a unified 3D VAE to extract visual information for images and videos instead of using separate encoders for two tasks. In the experiment, the authors compare with several famous unified model to show their model is better than those sota models. They also ablate on the fusion mechaism in the 3D VAE design.

**Questions:**

1. How does flow matching compare to ddpm or autoregressive to optimize the visual generation in the unified model?
2. What is the advantage of 3D VAE compared to other unified tokenizers, for example, UniTok (https://arxiv.org/abs/2502.20321), tokenflow (https://arxiv.org/pdf/2412.03069) and UniToken(https://arxiv.org/pdf/2504.04423)?
3. OmniCorpus is a large interrleave datasets, but its data quality is not very high (for example image and text are not very related). What kind of filtering is applied to get 12m examples?
4. Do you use the same image-text data for image understanding task and image generation task in stage 1?
5. I am curious about the ocr or doc understanding ability of unified model. what is the performance of unified model on textvqa, chartqa, docvqa, infovqa datasets?

**Ethical Concerns:**

["NO or VERY MINOR ethics concerns only"]

**Final Justification:**

The author's rebuttal has addressed all my concerns. I raised my score to accept.

**Limitations:**

yes

**Quality:**

3

**Strengths And Weaknesses:**

Strength:
1. The presentation is good and the paper is easy to follow.
2. The model is getting good performance compared to emu3, janus and show-o, which are famous unified multimodal models.
3. The paper proposes 3D VAE to jointly model the visual information for visual generation and understanding, which is novel.
4. It is a good trial to use flow matching for visual generation in the unified model, whch is novel based on the previous methods, but share some common grounds with blip3-o (another co-current work)

Weakness:
1. Some design choices are not explained clearly.
       (a) why uses flow matching rather than ddpm or autoregressive for generation?
       (b) how does 3D VAE encoder compare to other unified tokenizers?
       (c) Data filtering and curvation are not explained in details. (See Questions below)
2. Related Work session is not sufficient which makes it harder to understand the difference and novelty compared to previous works.

---

> ### Author Rebuttal · Authors · 2025-07-31
>
> **1. Why use flow matching rather than DDPM or autoregressive for generation?**
>
> We choose flow matching over DDPM because numerous studies [1,2] have demonstrated that flow matching converges faster and achieves better performance than DDPM. Additionally, flow matching has become the mainstream modeling paradigm for video generation [2,3,4] compared to pure autoregressive modeling. To further support this, we supplement the experimental results showing that flow matching converges more rapidly than DDPM on ImageNet.
> | Loss | Iterations | FID-5K ↓| IS ↑ |
> | :---        |    :----:   |:----:   |:----:   |
> | DDPM     |  100k | 16.47  | 269.41 |
> | Flow Matching     | 100k | 15.33 |276.17|
>
> Based on these observations and experiments, we adopt flow matching for image and video generation.
>
> [1] SD3: Scaling Rectified Flow Transformers for High-Resolution Image Synthesis. 2024.
> [2] Goku: Flow Based Video Generative Foundation Models, CVPR 2025.
> [3] HunyuanVideo: A Systematic Framework For Large Video Generative Models. 2024.
> [4] Wan: Open and Advanced Large-Scale Video Generative Models. 2024.
>
> **2. The advantages and comparisons of 3D VAE encoder over other unified tokenizers.**
>
> The inherent advantages of the 3D VAE lie in its flexible support for both image and video encoding and decoding, as well as its ability to achieve high-quality reconstruction for both modalities. To push the upper bound of unified multimodal models in image and video generation, we adopt the 3D VAE—a technique widely used in state-of-the-art commercial and open-source video generation models such as WAN2.1.
>
> While UniTok, UniToken, and TokenFlow offer elegant unified discrete token representations for image modalities, they do not extend their solutions to video modalities. Moreover, discrete token-based video tokenizers often lag behind their continuous counterparts, limiting the potential performance ceiling.
>
> Based on these considerations, we propose constructing unified visual representations within the 3D VAE latent space, providing a robust and effective approach to unifying both image and video modalities.
>
> Thanks for pointing it out. We’d like to add these discussions to the related work section of our paper.
>
> We also supplement our models’ evaluation on video generation and understanding benchmarks below (**We omit the reference information and only select some of the metrics to present due to the limited space).
>
> | Models  | Total Params. | Total | Quality Score    | Semantic Score    | Subject Consistency    | Background Consistency    | Temporal Flickering    |  Motion Smoothness    | Dynamic Degree    |  Aesthetic Quality    | Imaging Quality    | Object Class    | Human Action    |
> |-------------------------|:---------:|:-----:|:-----:|:-----:|:-----:|:-----:|:-----:|:-----:|:-----:|:-----:|:-----:|:-----:|:-----:|
> | Video generation only models|
> | ModelScope  | 1.7B      | **75.75** | 78.05 | 66.54 | 89.87 | 95.29 | 98.28 | 95.79 | 66.39 | 52.06 | 58.57 | 82.25 | 92.40 |
> | CogVideoX | 5B     | **81.61** | 82.75 | 77.04 | 96.23 | 96.52 | 98.66 | 96.92 | 70.97 | 61.98 | 62.90 | 85.23 | 99.40 |
> | Kling      | -      | **81.85** | 83.39 | 75.68 | 98.33 | 97.60 | 99.30 | 99.40 | 46.94 | 61.21 | 65.62 | 87.24| 93.40 |
> | Step-Video-T2V| 30B  | **81.83** | 84.46 | 71.28 | 98.05 | 97.67 | 99.40 | 99.08 | 53.06 | 61.23 | 70.63 | 80.56 | 94.00 |
> | Gen-3       | -      | **82.32** | 84.11 | 75.17 | 97.10 | 96.62 | 98.61 | 99.23 | 60.14 | 63.34 | 66.82 | 87.81| 96.40 |
> | Unified multimodal models|
> | Emu3      | 8B     | **80.96** | -     | -     | 95.32 | 97.69 | -     | 98.93 | 79.27 | 59.64 | -     | 86.17 | 77.71 |
> | VILA-U  | 7B     | **74.01** |  -    |  -    |  -    |  -    |  -    |  -    |  -    |  -    |  -    |  -    |  -    |
> | HaploOmni | 9B     | **78.10** |  -    |  -    | 96.40 | 97.60 |  -    | 96.80 | 65.30 |  -    |  -    |  -    |  -    |
> | **Ours**  | 2B     | **81.34** | 82.10 | 78.31 | 97.28 | 96.78 | 97.68 | 98.25 | 40.83 | 65.15 | 67.06 | 94.81 | 95.20 |
>
>
> | Models                         | I2V Subject | I2V Background | Camera Motion | Subject Consistency | Background Consistency | Temporal Flickering | Motion Smoothness | Dynamic Degree | Aesthetic Quality | Imaging Quality |
> |-------------------------------|:-----------:|:--------------:|:-------------:|:-------------------:|:----------------------:|:-------------------:|:----------------:|:--------------:|:-----------------:|:---------------:|
> | Video generation only models|
> | ConsistI2V                    | 94.69       | 94.57          | 33.60         | 95.27               | 98.28                  | 97.56               | 97.38            | 18.62          | 59.00             | 66.92           |
> | VideoCrafter-I2V          | 90.97       | 90.51          | 33.58         | 97.86               | 98.79                  | 98.19               | 98.00            | 22.60          | 60.78             | 71.68           |
> | SVD-XT-1.1                                 | 97.51       | 97.62          | -             | 95.42               | 96.77                  | 99.17               | 98.12            | 43.17          | 60.23             | 70.23           |
> | Unified multimodal models|
> | **Ours**                                | 96.94       | 98.83          | 28.41         | 93.83               | 97.45                  | -                   | 97.76            | 25.85          | 61.92             | 69.87           |
>
>
> | Models     | Params. |ActivityNet-QA |NextQA|PercepTest|VideoMME wo/w-subs|
> | :---        |    :----:   |    :----:  |   :----:  |:----:  |:----:  |
> |Understanding only models	|
> | LLaVA-N-Video      | 32B |54.3  | 77.3 | 59.4 | 60.2/63.0 |
> | LLaVA-OneVision      | 0.5B   | 50.5 | 57.2 | 49.2| 44.0/43.5|
> | LLaVA-OneVision      |  7B  | 56.6 | 79.4 | 57.1 | 58.2/61.5 |
> | VideoLLaMA2      |  7B  | 50.2 | - | 51.4|47.9/50.3 |
> | Unified multimodal models|
> | **Ours**    | 1.5B  | 52.7 | 72.1 | 56.1|48.0/51.6 |
> | **Ours**    | 7B  | 56.4 | 79.0 | 57.0|57.4/60.9 |
>
>
> **3. Data filtering strategies on interleaved datasets.**
>
> We did simple data filtering only based on the image resolution, and whether the image link is legacy to filter out those sequences with missing image files. However, we unfortunately find that the data quality is still relatively low, so we add two datasets, including CoMM and VIST as high-quality downstream data to further fine-tune the model. We will add these details to our paper.
>
> **4. Related work section.**
>
> Thanks for your suggestion. We plan to add more descriptive comparisons between our method and other related works and add our response to question 2 to the related work section.
>
> **5. Do you use the same image-text data for image understanding task and image generation task in stage 1?**
>
> Yes. In stage 1, we sample from the same data source to construct sequences for both image understanding and generation learning.
>
> **6. OCR and Doc understanding ability. What is the performance of unified model on textvqa, chartqa, docvqa, infovqa datasets?**
>
> Since our model was trained using a fixed image resolution of 432×432 (resulting in 729 tokens per image), we make a fair comparison between LLaVA-OneVision and our models under the same inference settings.
>
> | Models | # Image tokens | ChartQA| DocVQA_val |InfoVQA_val|TextVQA_val|
> | :---        |    :----:   |:----:   |:----:   |:----:   |:----:   |
> | LLaVA-OneVision-7B     |  729 | 56.24  | 62.71 |39.59|66.19|
> | Ours-7B    | 729| 48.00|59.34|42.31|62.92|
>
> As shown above, our model achieves comparable performance to LLaVA-OV-7B on DocVQA, InfoVQA, and TextVQA validation sets. However, it underperforms on the ChartQA task. We believe this is due to insufficient training data for chart types during our semantic layer distillation, which limits the model’s ability to maintain such fine-grained information effectively. We are confident that incorporating more OCR and document data into the distillation process will further enhance the unified model’s OCR and document understanding capabilities.
>
>
> Additionally, an interesting observation is that although our model was trained with a fixed image resolution, increasing the number of image tokens using the AnyRes strategy during inference consistently improves performance.
>
> | Models | # Max Image tokens | ChartQA| DocVQA_val |InfoVQA_val|TextVQA_val|
> | :---        |    :----:   |:----:   |:----:   |:----:   |:----:   |
> | Ours-7B    | 729| 48.00|59.34|42.31|62.92|
> | Ours-7B    | 5*729| 66.92|77.26|45.80|71.54|
>
> These results demonstrate that employing high-resolution images during training would potentially further improve the OCR and document understanding capabilities of our models.

---

### Official Review · Reviewer_H9zV · 2025-07-01

**Clarity:** 4
**Significance:** 3
**Originality:** 3
**Rating:** 5
**Confidence:** 4

**Summary:**

This paper presents Show-o2 that leverage autoregressive modeling and flow matching to improve native unified multimodal models. The authors introduce a novel visual representation scheme based on a 3D causal VAE to combine high-level semantic features with low-level pixel details. A key contribution is a two-stage training recipe designed to efficiently impart visual generation capabilities to the LLM without degrading its pre-trained language abilities or requiring massive new text corpora. The paper provides several experiments showing that Show-o2 variants achieve state-of-the-art or highly competitive performance on a wide range of benchmarks for multimodal understanding and visual generation.

**Questions:**

* On Data: In Section 3.2, the authors state, *However, this process often relies on web-scale, high-quality text corpora, which are prohibitively expensive to collect. Consequently, the lack of such resources can lead to a degradation in language knowledge and modeling performance* However, in the previous version of the paper, **Show-o, the authors incorporated the RefineWeb text corpus, which is open-source and not as expensive.** Why was it feasible to include this corpus in Show-o, but not in Show-o2, based on the justification provided?

* On the Model: The authors do not provide a clear explanation or ablation study for the decision to use flow matching as the generation method. The predecessor, **Show-o, used discrete diffusion modeling**, and such a significant architectural shift requires justification, preferably supported by ablation studies or other forms of evidence.

* On the Significance of Unified Native Multimodal Learning: While the authors have conducted extensive experiments comparing their model to previous unified models, achieving state-of-the-art (SOTA) is not a fundamental contribution in itself, as SOTA results are inevitably superseded by larger models trained on larger-scale, higher-quality data. For an academic conference paper, I am more interested in the **complementary or synergistic benefits** that generative multimodal pretraining brings to both understanding and generation tasks. **For understanding tasks, does generative visual training lead to improvements in the model's grasp of visual details, spatial reasoning, or spatio-temporal consistency?** For generation tasks, does effective long-form image-text training enhance the ability to follow complex instructions?
Regarding the understanding tasks, **the authors used data from LLaVA-OneVision but compared their model against LLaVA-v1.5.** I understand that comparing against LLaVA-OneVision's own metrics might make Show-o2's performance appear less favorable, but as a reviewer, I am more eager to see if Show-o2 is relatively balanced on vision-centric tasks like spatial understanding compared to the standard ViT+MLP+LLM paradigm. Unfortunately, I did not see the authors clearly and explicitly demonstrate such results.

* On Figure 2: The capabilities of Show-o2 could be more effectively highlighted in Figure 2 if some qualitative comparisons with other models were included.

**Ethical Concerns:**

["NO or VERY MINOR ethics concerns only"]

**Final Justification:**

The rebuttal has addressed my questions, and my final decision is "accept".

**Limitations:**

As shown above

**Paper Formatting Concerns:**

-

**Quality:**

3

**Strengths And Weaknesses:**

**Strengths**
* Architecture Design: The dual-path fusion mechanism  is an intelligent way to create a visual representation that serves the distinct needs of understanding (requiring semantics) and generation (requiring fine-grained detail). The native integration of autoregressive modeling with flow matching  within a single transformer is a sophisticated and effective approach.
* Evaluation: The authors have been exceptionally thorough, testing their models against a vast array of SOTA methods on numerous, diverse benchmarks for both understanding and generation.
* Clarity: The paper is, for the most part, clearly written and well-organized.

**Weaknesses**
* The video understanding section lacks comparison with state-of-the-art models.
* A mutually complementary or synergistic relationship between the model's generation and understanding capabilities is not clearly demonstrated.

---

> ### Author Rebuttal · Authors · 2025-07-31
>
> **1. Video benchmarks.**
>
> We supplement the quantitative evaluation of video understanding as follows, as well as the text-to-video generation on VBench:
>
> | Models     | Params. |ActivityNet-QA |NextQA|PercepTest|VideoMME wo/w-subs|
> | :---        |    :----:   |    :----:  |   :----:  |:----:  |:----:  |
> |Understanding only models	|
> | LLaVA-N-Video      | 32B |54.3  | 77.3 | 59.4 | 60.2/63.0 |
> | LLaVA-OneVision      | 0.5B   | 50.5 | 57.2 | 49.2| 44.0/43.5|
> | LLaVA-OneVision      |  7B  | 56.6 | 79.4 | 57.1 | 58.2/61.5 |
> | VideoLLaMA2      |  7B  | 50.2 | - | 51.4|47.9/50.3 |
> |Unified multimodal models|
> | **Ours**    | 1.5B  | 52.7 | 72.1 | 56.1|48.0/51.6 |
> | **Ours**    | 7B  | 56.4 | 79.0 | 57.0|57.4/60.9 |
>
> We follow the settings of LLaVA-OneVision to enhance our models with video understanding capabilities. As shown above, our model demonstrates comparable performance to LLaVA-OneVision and outperforms VideoLLaMA2 as well as the much larger LLaVA-N-Video-32B on several metrics.
>
> | Models  | Total Params. | Total | Quality Score    | Semantic Score    | Subject Consistency    | Background Consistency    | Temporal Flickering    |  Motion Smoothness    | Dynamic Degree    |  Aesthetic Quality    | Imaging Quality    | Object Class    | Human Action    |
> |-------------------------|:---------:|:-----:|:-----:|:-----:|:-----:|:-----:|:-----:|:-----:|:-----:|:-----:|:-----:|:-----:|:-----:|
> | Video generation only models|
> | ModelScope  | 1.7B      | **75.75** | 78.05 | 66.54 | 89.87 | 95.29 | 98.28 | 95.79 | 66.39 | 52.06 | 58.57 | 82.25 | 92.40 |
> | CogVideoX | 5B     | **81.61** | 82.75 | 77.04 | 96.23 | 96.52 | 98.66 | 96.92 | 70.97 | 61.98 | 62.90 | 85.23 | 99.40 |
> | Kling       | -      | **81.85** | 83.39 | 75.68 | 98.33 | 97.60 | 99.30 | 99.40 | 46.94 | 61.21 | 65.62 | 87.24| **93.40** |
> | Step-Video-T2V | 30B  | **81.83** | 84.46 | 71.28 | 98.05 | 97.67 | 99.40 | 99.08 | 53.06 | 61.23 | 70.63 | 80.56 | 94.00 |
> | Gen-3       | -      | **82.32** | 84.11 | 75.17 | 97.10 | 96.62 | 98.61 | 99.23 | 60.14 | 63.34 | 66.82 | 87.81| 96.40 |
> | Unified multimodal models|
> | Emu3      | 8B     | **80.96** | -     | -     | 95.32 | 97.69 | -     | 98.93 | 79.27 | 59.64 | -     | 86.17 | 77.71 |
> | HaploOmni | 9B     | **78.10** |  -    |  -    | 96.40 | 97.60 |  -    | 96.80 | 65.30 |  -    |  -    |  -    |  -    |
> | **Ours**  | 2B     | **81.34** | 82.10 | 78.31 | 97.28 | 96.78 | 97.68 | 98.25 | 40.83 | 65.15 | 67.06 | 94.81 | 95.20 |
>
>
> | Models                         | I2V Subject | I2V Background | Camera Motion | Subject Consistency | Background Consistency | Temporal Flickering | Motion Smoothness | Dynamic Degree | Aesthetic Quality | Imaging Quality |
> |-------------------------------|:-----------:|:--------------:|:-------------:|:-------------------:|:----------------------:|:-------------------:|:----------------:|:--------------:|:-----------------:|:---------------:|
> | Video generation only models|
> | ConsistI2V                    | 94.69       | 94.57          | 33.60         | 95.27               | 98.28                  | 97.56               | 97.38            | 18.62          | 59.00             | 66.92           |
> | VideoCrafter-I2V          | 90.97       | 90.51          | 33.58         | 97.86               | 98.79                  | 98.19               | 98.00            | 22.60          | 60.78             | 71.68           |
> | SVD-XT-1.1                                 | 97.51       | 97.62          | -             | 95.42               | 96.77                  | 99.17               | 98.12            | 43.17          | 60.23             | 70.23           |
> | Unified multimodal models|
> | **Ours**                                | 96.94       | 98.83          | 28.41         | 93.83               | 97.45                  | -                   | 97.76            | 25.85          | 61.92             | 69.87           |
>
>
>
> As shown above, our models can achieve competitive video generation performance compared to specialized video generation methods, and achieve better performance than unified multimodal models like Emu3 and HaploOmni.
>
> **2. Regarding text corpus.**
>
> Publicly available text corpora, such as RefinedWeb, tend to be of relatively low quality and insufficient for reproducing strong large language models such as the Qwen-2.5, which rely on extremely high-quality and diverse proprietary text data that is prohibitively expensive for most institutions to collect. Consequently, unified co-training with image-text pairs and such low-quality publicly available text corpora may potentially degrade the original language knowledge. We provide the evaluation results on text-only benchmarks using the lm-evaluation-harness tool [1] below for demonstration.
>
> | Models |Training Recipe | MMLU |GPQA|GSM8K|HumanEval|
> | :---        |    :----:   |:----:   |    :----:  |   :----:  |:----:  |
> | Qwen2.5-1.5B-Instruct     | - |60.20±0.39  |  28.12±2.13|51.86±1.38 | 35.37±3.74|
> | Ours-1.5B|one stage full-parameter-co-training with image-text and text-only data     |  28.25±0.38  | 25.00±2.05 | 4.55±0.57 | 3.05±1.35|
> | Ours-1.5B| our two-stage training  | 56.75±1.37   | 29.24±2.15| 49.43±1.38| 35.54±3.70|
> | Qwen2.5-7B-Instruct     | - |71.75±0.36 |  32.37±2.21|82.49±1.05 | 65.24±3.73|
> | Ours-7B|one stage full-parameter-co-training with image-text and text-only data  |  28.43±0.21  | 26.34±2.08 | 1.52±0.34 | 4.01±1.25|
> | Ours-7B| our two-stage training  | 70.73±0.36   | 31.47±2.22 | 75.28±1.19| 70.73±3.56|
>
> Directly co-training with all model parameters in one stage using both image-text pairs and textual data, such as RefinedWeb, leads to significant performance degradation, underscoring the necessity of our two-stage training approach when high-quality corpora are unavailable.
>
> [1] github repository: lm-evaluation-harness
>
> **3. Synergies.**
>
> Thank you for sharing your valuable comments. We are also interested in exploring the mutual effects within unified learning. Our observations and insights can be categorized into three aspects:
>
> i) Representation Level: Typically, multimodal understanding and visual generation rely on semantic and low-level (fine-grained) features, respectively. Based on our pilot study, we found that constructing unified visual representations by combining these two types of features leads to improvements in both understanding and generation, as demonstrated in the tables below.
>
> | Semantic Layers     | Projector | MME-p↑|GQA↑|POPE↑|
> | :---        |    :----:   |    :----:  |   :----:  |:----:  |
> | √      |  × | 1164.7 | 56.2 | 82.6 |
> | √      | √  | 1187.8 | 57.6 | 82.6 |
>
> | Semantic Layers     |Projector | FID-5K↓|
> | :---        |    :----:   |:----:   |
> | ×     |  √ | 21.8 |
> | √      | √  | 20.5 |
>
> ii) Objective Level: We conducted controlled experiments to investigate whether combining understanding and generation objectives leads to conflicts or mutual benefits. Our findings indicate no conflicts in multimodal understanding, as the model trained with unified learning achieved nearly identical performance on understanding benchmarks compared to the model trained solely with the understanding loss. Encouragingly, for visual generation, we observed improvements on the Dense Prompt Graph Benchmark (DPG-Bench), particularly in metrics related to color and relational accuracy, as shown below.
>
> | Understanding loss | Generation loss | Color↑ |Spatial relation↑|Non-spatial relation↑ |
> |:---------------:|:---------:|:-------:|:-------:|:-------:|
> | √               | ×         | 91.03    |90.63|86.09|
> | √               | √         | 93.57| 92.33| 89.38|
>
>
> iii) Complex Task Level: With the advancement of multimodal chain-of-thought techniques, there is a growing demand for generating intermediate or auxiliary images to assist in solving complex problems such as mazes or mathematical reasoning. We believe that the native unified multimodal model is well-suited to meet these challenges.
>
> We believe our models provide a strong baseline, and by open-sourcing our code, we hope to inspire further exploration in these three areas to continuously explore the mutual benefits and push the boundaries of unified multimodal models.
>
>
> **4. Comparison to LLaVA-OneVision.**
>
> Thanks for your suggestion. We supplement our latest comparison below:
> | Model                      | Params | MME ↑ | MMB ↑ | MMMU ↑ | MMStar ↑ | AI2D ↑ |
> |----------------------------|:------:|:------:|:----:|:----:|:----:|:----:|
> | LLaVA-OneVision  | 7B     | 1580.0 | 80.8 | 48.8 | 61.7| 81.4 |
> | Ours | 7B |  1620.5  |   79.3  |  48.9  |  56.6  |  78.6  |
>
> As shown above, for most multimodal understanding metrics, our model can achieve competitive performance with LLaVA-OneVision. We will add these results to the main table.
>
> **5. Flow matching vs (discrete) diffusion.**
>
> We choose flow matching over (discrete) diffusion because numerous studies [1,2] have demonstrated that flow matching converges faster and achieves better performance than diffusion. Additionally, flow matching has become the mainstream modeling paradigm for video generation [2,3,4]. Furthermore, continuous 3D VAE has better video reconstruction capability than discrete ones. So, we adopt flow matching with a continuous 3D causal VAE. We provide proof-of-concept experimental results showing that flow matching converges faster than diffusion on ImageNet.
> | Loss | Iterations | FID-5K ↓| IS ↑ |
> | :---        |    :----:   |:----:   |:----:   |
> | DDPM     |  100k | 16.47  | 269.41 |
> | Flow Matching     | 100k | 15.33 |276.17|
>
> [1] SD3: Scaling Rectified Flow Transformers for High-Resolution Image Synthesis
> [2] Goku: Flow Based Video Generative Foundation Models
> [3] HunyuanVideo: A Systematic Framework For Large Video Generative Models
> [4] Wan: Open and Advanced Large-Scale Video Generative Models
>
> **6. Visual comparisons.**
>
> Thanks for your suggestion. We will include visual comparisons among different unified multimodal models.

---

> > ### Comment · Reviewer_H9zV · 2025-08-07
> >
> > Thank you very much for the comprehensive experiments and targeted responses; I am very satisfied. I just have one more question I'd like to confirm:
> >
> > In the experimental results presented in the section "2. Regarding text corpus," the model "show-o2" undergoes "one stage full-parameter co-training with image-text and text-only data," which leads to a significant drop in metrics on the text evaluation set. What is the impact of this on the "multimodal understanding" task? I look forward to seeing the author's experimental results.

---

> > > ### Author Response · Authors · 2025-08-07
> > >
> > > We are very delighted to hear that you are very satisfied with our responses. **We just conducted an evaluation of the 1.5B model**  trained under the "one-stage full-parameter co-training with image-text and text-only data" setting on various multimodal understanding benchmarks. The results are shown below:
> > >
> > > | Training Recipe    | MME ↑| GQA↑| SEED ↑ |MMB↑| MMMU ↑ |MMStar ↑ |AI2D ↑|
> > > | :---        |    :----:   |    :----:  |   :----:  |:----:  |:----:  |:----:  |:----:  |
> > > | One-stage co-training     | 1347.2 | 56.7 | 62.8| 60.4|33.1|39.2|59.8|
> > > | Our two-stage training     | 1450.9 |60.0 |65.6 |67.4 |37.1 |43.4| 69.0|
> > >
> > > As shown above, the model trained with the one-stage co-training approach experiences a consistent performance drop, underperforming compared to the model trained with our two-stage training, which better preserves language knowledge. This outcome is expected, as a base model with better textual knowledge may more effectively serve for multimodal understanding.
> > >
> > > We will **add these experimental results of both 1.5B and 7B models** regarding language knowledge to the appendix of our paper to serve as a reference for future research.

---

> > > > ### Comment · Reviewer_H9zV · 2025-08-08
> > > >
> > > > The author's response has addressed my questions. If the experimental results from the rebuttal were incorporated into the paper, it would significantly aid readers in understanding the "show-o" series of works. I recommend accepting this paper for publication.

---

> > > > > ### Author Response · Authors · 2025-08-08
> > > > >
> > > > > Thank you very much for your responsiveness and positive feedback! We’re delighted that our responses have addressed your questions, and we will incorporate these supplementary experimental results to further enhance the comprehensiveness of our paper.

---

> > > > > > ### Author Response · Authors · 2025-08-08
> > > > > >
> > > > > > Thank you very much again for recommending acceptance of our paper. We would appreciate it very much if you could set your final rating as “accept.” Have a nice day!

---

> ### Author Response · Authors · 2025-07-31
> **Table revision.**
>
> |   Understanding loss   | Generation loss | Color↑|Spatial relation↑|Non-spatial relation↑|
> | :---        |    :----:   |    :----:  |   :----:  |:----:  |
> |    ×  |  √  | 91.03 | 90.63	|86.09|
> | √      | √  | 93.57|	92.33|	89.38|
>
> Apologies for the earlier mistakes. We have revised the table as shown above, indicating that improvements in metrics (DPG-Bench) related to color and relation accuracy have been observed in unified learning with the understanding loss.

---

### Official Review · Reviewer_RCaP · 2025-07-02

**Clarity:** 2
**Significance:** 2
**Originality:** 2
**Rating:** 3
**Confidence:** 5

**Summary:**

This paper proposes a native unified multimodal model designed to perform both multimodal understanding and generation within a single framework. The architecture is built upon a 3D causal VAE for unified visual representation, a large language model backbone, and a hybrid decoding scheme using an autoregressive head for text and a flow matching head for visuals. Key ideas include a dual-path mechanism for fusing high-level semantics and low-level details, and a two-stage training recipe to integrate generative capabilities without degrading the LLM's language proficiency. The authors claim state-of-the-art (SOTA) performance across a wide range of benchmarks.

**Questions:**

1. Regarding the dual-path fusion, could you provide an ablation study where you train the model with only the semantic path (S(·)) and only the projector path (P(·))? Please report performance on both understanding (e.g., GQA) and generation (e.g., FID) tasks to demonstrate the necessity and distinct roles of each path.
2. To substantiate the claim that the two-stage training preserves language ability, can you report the performance of your 7B model on a text-only benchmark like MMLU, comparing the results after your proposed two-stage training versus a single-stage joint training approach?

**Ethical Concerns:**

["Major Concern: Data privacy, copyright, and consent"]

**Final Justification:**

After reading the authors' rebuttal, many of my confusions have indeed been well addressed. However, I am still concerned that this work is essentially a combination of several existing techniques, making it difficult to deliver novel insights, and it remains unclear to me which components are truly effective.

**Limitations:**

Yes

**Quality:**

2

**Strengths And Weaknesses:**

### Strengths
+ Ambitious Goal: The work tackles the important and challenging problem of creating a single, native model for unified multimodal understanding and generation across text, image, and video.
+ Strong Empirical Performance (on image tasks): The model demonstrates competitive or SOTA results on several well-established image-based understanding and generation benchmarks (e.g., MME, DPG-Bench), indicating the potential of the proposed architectural design.

### Weaknesses
- Incremental Novelty: The core method is primarily a complex integration of existing techniques (3D Causal VAE from [67], Flow Matching from [41, 45], omni-attention from [75, 80]). The main novel component, the dual-path fusion, is a relatively straightforward feature combination. The paper lacks a core conceptual breakthrough and reads more like a strong engineering effort than a fundamental research advance.
- Insufficient Justification for Key Design Choices: The paper fails to provide adequate ablation studies to justify its core architectural decisions.
- Flow Matching vs. Diffusion: There is no comparison or discussion justifying the choice of Flow Matching over the more prevalent Diffusion Models for generation, making this choice seem arbitrary.
- Two-Stage Training: The claim that the two-stage recipe "retains language knowledge" is unsubstantiated. A critical missing experiment is a direct comparison against a single-stage, end-to-end trained model, evaluated on text-only benchmarks (e.g., MMLU) to quantify the alleged preservation of language capabilities.
- Significant Mismatch Between Claims and Evidence: The paper's claims, particularly regarding video and interleaved modalities, are not well-supported by the evidence provided.
- Unverified Video Capabilities: Video support is a key selling point of the architecture. However, the paper presents no quantitative results on any standard video benchmark.
- Weak Evidence for Interleaved Generation: Similarly, capabilities for mixed-modality generation are only supported by two qualitative examples in Fig. 2, which is insufficient for a rigorous evaluation.
- Lack of Experimental Rigor: SOTA claims are undermined by a lack of statistical analysis. No error bars or standard deviations are reported, making it difficult to assess the significance of marginal improvements over prior work.

---

> ### Author Rebuttal · Authors · 2025-07-31
>
> **1. Novelty.**
>
> We argue that our work presents significant innovations and contributions, which are also consistently acknowledged by the other three reviewers from different aspects: i) “use a 3D causal VAE to jointly model visual information for both generation and understanding, which is **novel**.” commented by **Reviewer sCtb**; ii) “The unified visual representation is **novel and intuitive**” demonstrated by **Reviewer hZso**; iii) “The dual-path fusion mechanism is **an intelligent way** to create a visual representation” noted by **Reviewer H9zV**.
>
> In addition, we argue that integrating 3D causal VAE, flow matching, and omni-attention is a logical exploration of more flexible architectural designs that unify a broader range of vision modalities, including both images and videos, in which video modalities remain relatively underexplored in the unified model literature. Further, our proposed two-stage training recipe alleviates the need for large-scale, high-quality text corpora, which are prohibitively expensive for most institutions to collect.
>
> While our work may not represent a groundbreaking conceptual breakthrough on the scale of GPT or Diffusion, it provides significant insights and strong experimental results that pave the way for future research in unifying text, images, and videos. We will open-source our training code and model weights to establish a reproducible platform and inspire further exploration in this area.
>
> **2. Ablation study on dual-path fusion.**
> We employ a 1B-parameter LLM as the base model to conduct an efficient ablation study. All experiments are carried out under consistent training settings and datasets. The experimental results are shown in Table 6 and presented below:
> | Semantic Layers     | Projector | MME-p↑|GQA↑|POPE↑|
> | :---        |    :----:   |    :----:  |   :----:  |:----:  |
> | √      |  × | 1164.7 | 56.2 | 82.6 |
> | √      | √  | 1187.8 | 57.6 | 82.6 |
>
> | Semantic Layers     |Projector | FID-5K↓|
> | :---        |    :----:   |:----:   |
> | ×     |  √ | 21.8 |
> | √      | √  | 20.5 |
>
> For multimodal understanding, the baseline model utilizes only semantic features (known to be more effective for understanding) extracted by the semantic layers, while for visual generation, the baseline relies solely on VAE features (recognized as crucial for generation) processed through the projector. As demonstrated above, combining these two types of features into a unified visual representation leads to improved performance across both multimodal understanding and generation metrics. This highlights the effectiveness and necessity of our proposed fusion strategy.
>
> **3. Flow matching vs DDPM.**
> We choose flow matching over DDPM because numerous studies [1,2] have demonstrated that flow matching converges faster and achieves better performance. Additionally, flow matching has become the mainstream modeling paradigm for video generation [2,3,4]. To further validate this, we present our proof-of-concept experimental results on the ImageNet dataset, which show that flow matching converges more rapidly than DDPM.
> | Loss | Iterations | FID-5K ↓| IS ↑ |
> | :---        |    :----:   |:----:   |:----:   |
> | DDPM     |  100k | 16.47  | 269.41 |
> | Flow Matching     | 100k | 15.33 |276.17|
>
> [1] SD3: Scaling Rectified Flow Transformers for High-Resolution Image Synthesis. 2024.
> [2] Goku: Flow Based Video Generative Foundation Models, CVPR 2025.
> [3] HunyuanVideo: A Systematic Framework For Large Video Generative Models. 2024.
> [4] Wan: Open and Advanced Large-Scale Video Generative Models. 2024.
>
>
> **4. Effectiveness of two-stage training.**
> We evaluate the Qwen2.5-1.5B-Instruct, Qwen2.5-7B-Instruct, the model co-trained with both image-text pairs and text-only data i.e., RefinedWeb, in single-stage joint training, and our own 1.5B and 7B models in two-stage training using the lm-evaluation-harness tool [1] under identical evaluation settings. The results are presented below.
>
> | Models |Training Recipe | MMLU |GPQA|GSM8K|HumanEval|
> | :---        |    :----:   |:----:   |    :----:  |   :----:  |:----:  |
> | Qwen2.5-7B-Instruct     | - |71.75±0.36 |  32.37±2.21|82.49±1.05 | 65.24±3.73|
> | Ours-7B|single-stage joint training  |  28.43±0.21  | 26.34±2.08 | 1.52±0.34 | 4.01±1.25|
> | Ours-7B| our two-stage training  | 70.73±0.36   | 31.47±2.22 | 75.28±1.19| 70.73±3.56|
>
> Our models trained in two-stage recipe effectively preserve language knowledge and achieve performance comparable to the original Qwen2.5 Instruct models. In contrast, directly end-to-end training in a single stage leads to significant performance degradation, underscoring the necessity of our two-stage training when high-quality corpora are unavailable.
>
> [1] github repository: lm-evaluation-harness
>
> **5. Video benchmarks.**
> We supplement the quantitative evaluation of text-to-video generation and image-to-video generation on the widely adopted VBench as follows, as well as the video understanding evaluation:
> (**We omit the reference information and only select some of the models and metrics due to the limited space. Complete ones are in our responses to the other reviewers.**)
>
> | Models  | Total Params. | Total | Quality Score    | Semantic Score    | Subject Consistency    | Background Consistency    | Temporal Flickering    |  Motion Smoothness    | Dynamic Degree    |  Aesthetic Quality    | Human Action    |
> |-------------------------|:---------:|:-----:|:-----:|:-----:|:-----:|:-----:|:-----:|:-----:|:-----:|:-----:|:-----:|
> | Video generation only models|
> | ModelScope  | 1.7B      | **75.75** | 78.05 | 66.54 | 89.87 | 95.29 | 98.28 | 95.79 | 66.39 | 52.06 | 92.40 |
> | CogVideoX | 5B     | **81.61** | 82.75 | 77.04 | 96.23 | 96.52 | 98.66 | 96.92 | 70.97 | 61.98 | 99.40 |
> | Kling       | -      | **81.85** | 83.39 | 75.68 | 98.33 | 97.60 | 99.30 | 99.40 | 46.94 | 61.21 | 93.40 |
> | Step-Video-T2V | 30B  | **81.83** | 84.46 | 71.28 | 98.05 | 97.67 | 99.40 | 99.08 | 53.06 | 61.23 |94.00 |
> | Unified multimodal models|
> | Emu3      | 8B     | **80.96** | -     | -     | 95.32 | 97.69 | -     | 98.93 | 79.27 | 59.64 | 77.71 |
> | HaploOmni | 9B     | **78.10** |  -    |  -    | 96.40 | 97.60 |  -    | 96.80 | 65.30 |  -    |  -    |
> | **Ours**  | 2B     | **81.34** | 82.10 | 78.31 | 97.28 | 96.78 | 97.68 | 98.25 | 40.83 | 65.15 | 95.20 |
>
>
> | Models                         | Subject | Background | Camera Motion | Subject Consistency | Background Consistency | Temporal Flickering | Motion Smoothness |
> |-------------------------------|:-----------:|:--------------:|:-------------:|:-------------------:|:----------------------:|:-------------------:|:----------------:|
> | Video generation only models|
> | VideoCrafter-I2V          | 90.97       | 90.51          | 33.58         | 97.86               | 98.79                  | 98.19               | 98.00            |
> | SVD-XT-1.1                                 | 97.51       | 97.62          | -             | 95.42               | 96.77                  | 99.17               | 98.12            |
> | Unified multimodal models|
> | **Ours**                                | 96.94       | 98.83          | 28.41         | 93.83               | 97.45                  | -                   | 97.76            |
>
>
> | Models     | Params. |ActivityNet-QA |NextQA|PercepTest|VideoMME wo/w-subs|
> | :---        |    :----:   |    :----:  |   :----:  |:----:  |:----:  |
> | Understanding only models|
> | LLaVA-N-Video      | 32B |54.3  | 77.3 | 59.4 | 60.2/63.0 |
> | LLaVA-OneVision      |  7B  | 56.6 | 79.4 | 57.1 | 58.2/61.5 |
> | VideoLLaMA2      |  7B  | 50.2 | - | 51.4|47.9/50.3 |
> | Unified multimodal models|
> | **Ours**    | 7B  | 56.4 | 79.0 | 57.0|57.4/60.9 |
>
>
> As shown above, our models can achieve competitive video generation performance compared to specialized video generation methods, and achieve performance better than unified multimodal models like Emu3. Besides, we also attach the video understanding performance. From the table, we can see that our models achieve competitive performance compared to those understanding only models in terms of ActivityNet-QA, Next-QA, Perceptual-Test, and Video-Meme metrics.
>
> **6. Interleaved generation evaluation.**
> As demonstrated by our experimental results, this work focuses more on demonstrating the flexible support across text, images, and videos. Due to the limited availability of high-quality interleaved image-text data and the incompleteness of existing evaluation benchmarks, we have not conducted comprehensive evaluations on mixed-modality generation. Instead, this work primarily serves to showcase the potential of our framework in handling mixed-modality generation. We plan to release the source code to the research community to facilitate further exploration in the future.
>
> **7. Error bars.**
> Due to time constraints, we conducted the second-stage experiments three times to estimate the standard deviation of our models' performance. The results are summarized below:
> | Models |MME ↑ |GQA↑| SEED ↑| MMB↑| MMMU ↑| MMStar ↑| AI2D ↑|
> | :---        |    :----:   |:----:   |    :----:  |   :----:  |:----:  |:----:  |:----:  |
> |Janus-Pro | 1567.1| 62.0 |72.1 |79.2 |41.0 |- |-|
> |Ours | 1621.2±1.6 | 63.1±0.05| 69.8±0.03 |79.3±0.07 |48.9±0.45 |56.6±0.015| 78.6±0.03|
>
> | Models |GenEval |DPG-Bench|
> | :---        |    :----:   |:----:   |
> |Janus-Pro |  0.73| 82.63 |
> |Ours | 0.73±0.0012| 85.02±0.13 |
>
> These results demonstrate that our model's performance is consistently stable across all metrics, and the observed standard deviations do not affect our advantage over previous state-of-the-art methods such as Janus-Pro. We will add these statistical analyses to the paper.
>
> **8. Ethical Concerns**
> We follow prior work in collecting data from publicly available datasets for research purposes. Our internal data undergoes some filtering to ensure the exclusion of unsafe content, user information, and copyright.

---

> > ### Author Response · Authors · 2025-08-05
> >
> > Dear Reviewer RCaP,
> >
> > Thank you once again for your valuable feedback, which has greatly contributed to improving our work. We have addressed all the questions you raised in your review and kindly invite you to review our responses. Please let us know if anything requires further clarification.
> >
> > Best regards,
> >
> > Authors of Paper 5552

---

> ### Author Response · Authors · 2025-08-07
>
> Dear Reviewer RCaP,
>
> Thank you for your thorough review and valuable feedback. We have carefully addressed your comments with extensive experimental results. As the discussion deadline approaches, please let us know if any points require further clarification.
>
> With sincere appreciation,
> Authors of Paper 5552

---

### Official Review · Reviewer_hZso · 2025-07-03

**Clarity:** 4
**Significance:** 3
**Originality:** 3
**Rating:** 4
**Confidence:** 4

**Summary:**

This paper proposes a unified multimodal model for both visual understanding and generation. The two key components for this unification are: 1) a fusion module that concatenates both semantic features (SigLIP) and low-level features (3D VAE) as visual input, and 2) a flow head to enable generation. The training involves two stages to retain language knowledge. Experiments are conducted on multimodal understanding and generation benchmarks, including both images and videos.

**Questions:**

1. More details are needed for the ablation of spatial(-temporal) fusion, how is the baseline set? Does the baseline also use both semantic and low-level VAE features?
2. How does the model perform on video generation evaluated quantitively?
3. It is not common for unified model to evaluate on pure text tasks. Since the paper makes the clain to retain language knowledge, I'd like to know how the language ability changes after multi-modal training.
4. This won't affect my ratings but still I wish to see some comparsion with some very recent work (after the submission ddl) like bagel[1], MetaQuery[2], UniWorld[3]

[1] Deng, Chaorui, et al. "Emerging properties in unified multimodal pretraining." arXiv preprint arXiv:2505.14683 (2025).
[2] Pan, Xichen, et al. "Transfer between modalities with metaqueries." arXiv preprint arXiv:2504.06256 (2025).
[3] Lin, Bin, et al. "Uniworld: High-resolution semantic encoders for unified visual understanding and generation." arXiv preprint arXiv:2506.03147 (2025).

**Ethical Concerns:**

["NO or VERY MINOR ethics concerns only"]

**Final Justification:**

I still recommend accept because it is a technically sound paper and most of my questions are addressed by the rebuttal.

However, most results reported in the rebuttal, are provided by an "updated model". This is not the original one submitted in the paper and should not be considered as the contribution of submitted paper for the sake of fairness. That's why I lower it to boardline accept.

Overall, the original paper, fails to:
1. demonstrate "unified" as no quantitative results of video generation are provided.
2. demonstrate "retaining language knowledge" as no text-only results are provided.

**Limitations:**

Yes

**Quality:**

3

**Strengths And Weaknesses:**

**Strengths**

1.  The unified visual representation is novel and intuitive. A key problem in unified models is that understanding requires high-level semantic features while generation requires low-level features. With the dual-path and channel-wise fusion, the model is fed both semantic and low-level features.
2.  The experiments, including the setup, are comprehensive and detailed. This is particularly good for the community for reproducibility.
3.  The performance is superior among native unified models. The model is also data-efficient overall (compared to Janus-Pro-7B).
4.  The writing is clear and easy to understand.

**Weaknesses**

1.  The performance gain from spatial fusion seems minimal based on Table 6. It is not detailed how the baseline without spatial fusion was conducted. Assuming it uses only the semantic feature, the improvement is not very impressive, especially for the image generation task in terms of FID-5K.
2.  Missing experimental results.
    1.  There are only qualitative results for video generation and no quantitative results. The visualizations of generated videos appear to be slow-motion or static, without much motion or camera movement. Most of the content consists of natural scenes like water, fire, and the sky. No results of human body motion are shown.
    2.  The paper claims to *retain language knowledge while simultaneously developing visual generation capabilities*, but no language-only benchmarks are evaluated. One may wonder how much of the model's language ability is preserved compared to the original Qwen2.5-1.5B-Instruct.

---

> ### Author Rebuttal · Authors · 2025-07-31
>
> **1. Missing details for the ablation study settings on spatial fusion.**
>
> Sorry for the missing details. We reorganized the table as follows for clarity:
>
> | Semantic Layers     | Projector | MME-p↑|GQA↑|POPE↑|
> | :---        |    :----:   |    :----:  |   :----:  |:----:  |
> | √      |  × | 1164.7 | 56.2 | 82.6 |
> | √      | √  | 1187.8 | 57.6 | 82.6 |
>
> | Semantic Layers     |Projector | FID-5K↓|
> | :---        |    :----:   |:----:   |
> | ×     |  √ | 21.8 |
> | √      | √  | 20.5 |
>
> For multimodal understanding, the baseline model utilizes only semantic features (known to be more effective for understanding) extracted by the semantic layers, while for visual generation, the baseline relies solely on VAE features (recognized as crucial for generation) processed through the projector. As demonstrated above, combining these two types of features into a unified visual representation leads to improved performance across both multimodal understanding and generation metrics.
>
> Our primary goal is to construct a unified visual representation. At a minimum, we expect not to degrade the original performance, and surprisingly, the combination yields mutually beneficial improvements, even if they are not very significant. I believe this observation is very valuable and will inspire further research to explore this property in greater depth.
>
> We will update Table 6 to avoid misunderstanding.
>
>
> **2. Video generation quantitative results.**
>
> We supplement the quantitative evaluation of text-to-video generation and image-to-video generation on the widely adopted VBench as follows, as well as the video understanding evaluation (**We omit the reference information and only select some of the metrics to present due to the limited space**):
>
> | Models  | Total Params. | Total | Quality Score    | Semantic Score    | Subject Consistency    | Background Consistency    | Temporal Flickering    |  Motion Smoothness    | Dynamic Degree    |  Aesthetic Quality    | Imaging Quality    | Object Class    | Human Action    |
> |-------------------------|:---------:|:-----:|:-----:|:-----:|:-----:|:-----:|:-----:|:-----:|:-----:|:-----:|:-----:|:-----:|:-----:|
> | Video generation only models|
> | ModelScope  | 1.7B      | **75.75** | 78.05 | 66.54 | 89.87 | 95.29 | 98.28 | 95.79 | 66.39 | 52.06 | 58.57 | 82.25 | **92.40** |
> | CogVideoX | 5B     | **81.61** | 82.75 | 77.04 | 96.23 | 96.52 | 98.66 | 96.92 | 70.97 | 61.98 | 62.90 | 85.23 | **99.40** |
> | Kling       | -      | **81.85** | 83.39 | 75.68 | 98.33 | 97.60 | 99.30 | 99.40 | 46.94 | 61.21 | 65.62 | 87.24| **93.40** |
> | Step-Video-T2V | 30B  | **81.83** | 84.46 | 71.28 | 98.05 | 97.67 | 99.40 | 99.08 | 53.06 | 61.23 | 70.63 | 80.56 | **94.00** |
> | Gen-3       | -      | **82.32** | 84.11 | 75.17 | 97.10 | 96.62 | 98.61 | 99.23 | 60.14 | 63.34 | 66.82 | 87.81| **96.40** |
> | Unified multimodal models|
> | Emu3      | 8B     | **80.96** | -     | -     | 95.32 | 97.69 | -     | 98.93 | 79.27 | 59.64 | -     | 86.17 | **77.71** |
> | VILA-U  | 7B     | **74.01** |  -    |  -    |  -    |  -    |  -    |  -    |  -    |  -    |  -    |  -    |  -    |
> | HaploOmni | 9B     | **78.10** |  -    |  -    | 96.40 | 97.60 |  -    | 96.80 | 65.30 |  -    |  -    |  -    |  -    |
> | **Ours**  | 2B     | **81.34** | 82.10 | 78.31 | 97.28 | 96.78 | 97.68 | 98.25 | 40.83 | 65.15 | 67.06 | 94.81 | **95.20** |
>
>
> | Models                         | I2V Subject | I2V Background | Camera Motion | Subject Consistency | Background Consistency | Temporal Flickering | Motion Smoothness | Dynamic Degree | Aesthetic Quality | Imaging Quality |
> |-------------------------------|:-----------:|:--------------:|:-------------:|:-------------------:|:----------------------:|:-------------------:|:----------------:|:--------------:|:-----------------:|:---------------:|
> | Video generation only models|
> | ConsistI2V                    | 94.69       | 94.57          | 33.60         | 95.27               | 98.28                  | 97.56               | 97.38            | 18.62          | 59.00             | 66.92           |
> | VideoCrafter-I2V          | 90.97       | 90.51          | 33.58         | 97.86               | 98.79                  | 98.19               | 98.00            | 22.60          | 60.78             | 71.68           |
> | SVD-XT-1.1                                 | 97.51       | 97.62          | -             | 95.42               | 96.77                  | 99.17               | 98.12            | 43.17          | 60.23             | 70.23           |
> | Unified multimodal models|
> | **Ours**                                | 96.94       | 98.83          | 28.41         | 93.83               | 97.45                  | -                   | 97.76            | 25.85          | 61.92             | 69.87           |
>
>
>
> | Models     | Params. |ActivityNet-QA |NextQA|PercepTest|VideoMME wo/w-subs|
> | :---        |    :----:   |    :----:  |   :----:  |:----:  |:----:  |
> |Understanding only models	|
> | LLaVA-N-Video      | 32B |54.3  | 77.3 | 59.4 | 60.2/63.0 |
> | LLaVA-OneVision      | 0.5B   | 50.5 | 57.2 | 49.2| 44.0/43.5|
> | LLaVA-OneVision      |  7B  | 56.6 | 79.4 | 57.1 | 58.2/61.5 |
> | VideoLLaMA2      |  7B  | 50.2 | - | 51.4|47.9/50.3 |
> | Unified multimodal models|
> | **Ours**    | 1.5B  | 52.7 | 72.1 | 56.1|48.0/51.6 |
> | **Ours**    | 7B  | 56.4 | 79.0 | 57.0|57.4/60.9 |
>
>
> As shown above, our models can achieve competitive video generation performance compared to specialized video generation models such as CogVideo-X and Gen-3, and achieve decent performance better than unified multimodal models like Emu3, VILA-U, and HaploOmni. Besides, following the LLaVA-OneVision training settings, we also attach the video understanding performance. From the table, we can see our models can achieve competitive performance compared to those understanding only models in terms of Activitynet-QA, Next-QA, PercepTest, and VideoMME metrics.
>
>
> **3. Camera motion and human body motion results.**
>
> The quality and motion dynamics of video generation heavily depend on the datasets used. In the submitted version, our model was fine-tuned on a relatively small set of publicly available data, which may have limited its performance. However, as shown in the tables above, our updated model achieves competitive results across multiple metrics, demonstrating the effectiveness of our proposed framework. Additionally, the above supplementary text-to-video evaluation includes a metric for ‘human action,’ which assesses the quality of human body action. According to the table, our updated model outperforms video generation models such as Step-T2V and Kling regarding this metric.
>
> **4. Language-only benchmarks.**
>
> We evaluated the Qwen2.5-1.5B-Instruct, Qwen2.5-7B-Instruct, the model co-trained with both image-text and text-only data i.e., RefinedWeb, in one stage full-training, and our own 1.5B and 7B models using the lm-evaluation-harness tool [1] under identical evaluation settings. The results are presented below.
>
>
> | Models |Training Recipe | MMLU |GPQA|GSM8K|HumanEval|
> | :---        |    :----:   |:----:   |    :----:  |   :----:  |:----:  |
> | Qwen2.5-1.5B-Instruct     | - |60.20±0.39  |  28.12±2.13|51.86±1.38 | 35.37±3.74|
> | Ours-1.5B|one stage full-parameter-co-training with image-text and text-only data     |  28.25±0.38  | 25.00±2.05 | 4.55±0.57 | 3.05±1.35|
> | Ours-1.5B| our two-stage training  | 56.75±1.37   | 29.24±2.15| 49.43±1.38| 35.54±3.70|
> | Qwen2.5-7B-Instruct     | - |71.75±0.36 |  32.37±2.21|82.49±1.05 | 65.24±3.73|
> | Ours-7B|one stage full-parameter-co-training with image-text and text-only data  |  28.43±0.21  | 26.34±2.08 | 1.52±0.34 | 4.01±1.25|
> | Ours-7B| our two-stage training  | 70.73±0.36   | 31.47±2.22 | 75.28±1.19| 70.73±3.56|
>
> It can be observed that our models effectively preserve language knowledge and achieve performance comparable to the original Qwen2.5-1.5B and Qwen2.5-7B Instruct models. In contrast, directly co-training in one stage using textual data such as RefinedWeb leads to significant performance degradation, underscoring the necessity of our two-stage training approach when high-quality corpora are unavailable.
>
> [1] github repository: lm-evaluation-harness
>
> **5. Comparison with very recent works.**
>
> Thanks for the suggestion. We attach the comparison below.
>
> | Models |Archecture|# Image-Text pairs|MME-p | MMMU |GenEval|DPG-Bench|
> | :---        |    :----:   |:----:   |    :----:  |   :----:  |:----:  |:----:  |
> |BAGEL|7B+7B MOT|1600M+500M|1687.0 |55.3|0.88|85.07|
> |MetaQuery|Qwen2.5-VL-7B+SANA-1.6B|# Data used in Qwen2.5-VL and SANA + 27.4M|1685.2|58.6|0.80|82.05|
> |UniWorld|Qwen2.5-VL-7B+FLUX-12B|# Data used in Qwen2.5-VL and FLUX + 2.7M|1685.2|58.6|0.84|81.38|
> |Ours updated mdoel | Qwen2.5-Instruct-7B| 66M + 9M|1620.5|48.9|0.76|86.14|
>
> One should notice that these recent works may leverage larger training datasets, bigger models, and powerful pretrained vision-language models such as Qwen2.5-VL, as well as pretrained text-to-image generation models like SANA and FLUX. In contrast, our model builds upon a purely language-pretrained models and shows promising capabilities and potential with a limited amount of training data.

---

> > ### Comment · Reviewer_hZso · 2025-08-05
> >
> > Thank you for your rebuttal. Most of my concerns are addressed. The paper overall looks solid and sound to me. However, most results in the rebuttal are obtained with an "updated model". This is a major update of the model after the submission deadline. For the sake of fairness, could the authors please clearly state which part of results in the rebuttal is obtained using an "updated model"? This will affect my final rating. I'm specially focusing on:
> >
> > 1. The video generation performance. This justifies the "unified" claim
> > 2. The text only performance. This justifies the "langauge retaining" claim
> > 3. The comparison with recent work. This justifies the "SoTA" claim

---

> ### Author Response · Authors · 2025-08-05
>
> Thank you for recognizing our paper as solid and sound, and I apologize for any misunderstandings.
>
> Regarding the video generation, we have added an additional 1.5 million high-quality samples along with the originally used OpenVid-1M dataset. The performance of text-to-video and image-to-video generation reported in our response reflects this update. **Compared to other video generation models that may utilize hundreds of millions of video samples, our dataset scale remains very small. However, this incorporation of high-quality data significantly demonstrates the effectiveness and potential of our proposed framework, helping to achieve competitive performance and enabling a relatively fair comparison with specialized video generation models that have larger model sizes and are trained on more extensive datasets.** We will include these results and discussions in our paper.
>
> We would like to clarify that the term "updated model" in the "Comparison with very recent work" table refers to our re-trained 7B model using cleaned-up code prepared for future open-source release. **This re-training follows the same training settings as the submission version.** As we found that there are minor variations in some metrics (such as MME 1623.5 --> 1620.5 and SEED-Bench 69.4 --> 69.8) when evaluating this model, compared to the original submission (but **the overall performance remains largely consistent**), we use the term "updated model" to avoid confusion regarding these slight differences in our response. **We confirm that “the updated model” means the re-training with the same settings rather than the other update, ensuring consistency with the submission version.**
>
> We directly evaluate the trained models to obtain text-only performance, which directly demonstrates the excellent capability of our models to retain language knowledge.
>
> I hope this clarifies the confusion.

---

### Author Response · Authors · 2025-08-09
**Thank You to the (Senior) Area Chairs and Reviewers: A Rebuttal Phase Recap**

We would like to sincerely thank the (Senior) Area Chairs and all reviewers for their time and effort in evaluating our paper and facilitating the discussion.

As the author-reviewer discussion period is coming to a close, we would like to summarize the current status and outcomes.

1.  Reviewer hZso identified several strengths of our work: (i) “The **unified visual representation is novel and intuitive**,” (ii) “The **experiments**, including the **setup**, are **comprehensive and detailed**,” (iii) “The model is also **data-efficient** overall,” and (iv) “The writing is **clear and easy** to understand.” After reviewing our responses, Reviewer hZso stated that **most concerns were addressed and acknowledged our work as overall solid and sound.** Regarding the “updated model” concern, we clarified that no changes were made to the model or training recipe; everything remains consistent with the original submission.

2.  Reviewer H9zV also recognized the merits of our work, noting: (i) “The dual-path fusion mechanism is **an intelligent way**,” (ii) “**Exceptionally thorough, testing their models against a vast array of SOTA methods,**” and (iii) the paper is, for the most part, **clearly written and well-organized**. During the discussion period, **Reviewer H9zV was very satisfied with our responses and recommended accepting our paper for publication**.

3.  Reviewer sCtb also highlighted several strengths of our work: (i) “**Achieving good performance**,” and (ii) “Proposes 3D VAE to jointly modeling visual information for both generation and understanding, which is **novel**.” **Reviewer sCtb did not further raise questions during the discussion period, and we are confident that our responses have adequately addressed his/her questions.**

4.  Reviewer RCaP raised some questions regarding the methodology and experiments. We have provided extensive supplementary experimental results to address these. However, **Reviewer RCaP did not participate in the discussion phase, and we have not received any further feedback. Since all the important questions raised by Reviewer RCaP overlap with those from the other three reviewers, and those reviewers have indicated that their concerns have been addressed or expressed satisfaction with our responses, we are confident that our responses have also adequately addressed Reviewer RCaP’s questions.**

We sincerely appreciate the insightful comments and constructive suggestions from all reviewers, which have greatly helped us improve our work. We remain committed to adding the experimental results in our responses to the final version of the paper to further enhance our contributions.

---

### Decision · Program_Chairs · 2025-09-17

**Decision:**

Accept (poster)

**Comment:**

This submission introduces a unified multimodal model that integrates flow matching for visual generation into an autoregressive LLMs, with components including a fusion of SigLIP semantic features and 3D VAE visual latents.

The final ratings are accept, accept, borderline accept, and borderline reject.
Although the reviewer RCaP acknowledged that the author's rebuttal addressed many of the confusions, the reviewer RCaP maintains the borderline reject and holds concerns about the technical novelty.
Reviewer hZso lowered the rating to borderline accept due to the concerns regarding the fairness of the "updated model" in the rebuttal.
The AC checked the response of the authors and thinks the explanation makes sense, i.e., "We confirm that “the updated model” means the re-training with the same settings rather than the other update, ensuring consistency with the submission version."
The other two reviewers gave clear accepts and recognized the contributions of the proposed model.

Overall, the paper is considered solid, with convincing responses and evaluation. Considering the good empirical performance, new insights and baselines to the community toward unified multimodal models, the AC recommends to accept this paper.
The authors should revise the paper according to the reviewers' comments.